# Multi-functional soft-bodied jellyfish-like swimming

Ziyu Ren [1,2], Wenqi Hu[1,2], Xiaoguang Dong[1] & Metin Sitti [1]

The functionalities of the untethered miniature swimming robots significantly decrease as the robot size becomes smaller, due to limitations of feasible miniaturized on-board components. Here we propose an untethered jellyfish-inspired soft millirobot that could realize multiple functionalities in moderate Reynolds number by producing diverse controlled fluidic flows around its body using its magnetic composite elastomer lappets, which are actuated by an external oscillating magnetic field. We particularly investigate the interaction between the robot's soft body and incurred fluidic flows due to the robot's body motion, and utilize such physical interaction to achieve different predation-inspired object manipulation tasks. The proposed lappet kinematics can inspire other existing jellyfish-like robots to achieve similar functionalities at the same length and time scale. Moreover, the robotic platform could be used to study the impacts of the morphology and kinematics changing in ephyra jellyfish.

---

[1] Physical Intelligence Department, Max Planck Institute for Intelligent Systems, 70569 Stuttgart, Germany. [2] These authors contributed equally: Ziyu Ren, Wenqi Hu. Correspondence and requests for materials should be addressed to M.S. (email: sitti@is.mpg.de)

 1

Untethered miniature swimming robots[1–8] are indispensable in biomedical and environmental monitoring and remediation applications. Although existing miniature swimming robots have shown interesting mobility, their advanced functionalities, such as object manipulation ability, significantly decrease as the robot size becomes smaller, due to limitations of their miniaturized on-board components[9]. To achieve the object manipulation function, microswimmers operating in the low Reynolds number ($Re$) regime have been proposed to incur controlled viscous fluidic flows to manipulate objects[10–20]. However, it is unclear whether such approach is applicable in the moderate $Re$ regime, where both inertial and viscous forces play critical roles[21].

In nature, *scyphomedusae* ephyra, the juvenile of the most widely distributed jellyfish, can smartly control the fluidic flow around their body to realize diverse functionalities, such as propulsion[22–24], predation[25–28], and mixing of the surrounding fluid[29], despite their simple body structure. Inspired by ephyra, we propose an untethered jellyfish-like soft millirobot, which could realize multiple functionalities by producing diverse controlled fluidic flows around its body using its lappets, which are actuated by magnetic composite elastomer and bent by remote magnetic fields. Using this experimental setup, we study five distinct swimming modes to particularly investigate the interaction between the robot's soft body and incurred fluidic flows due to the robot's body motion and utilize such physical interaction for predation-inspired object manipulation capability of the robot, in addition to the robot's swimming propulsion, which has been the only focus of previous jellyfish-like robot studies[1,2,5,30]. The proposed soft robot's different lappet motion kinematics are used to conduct four different robotic tasks: selectively trap and transport objects of two different sizes, burrow into granular media consisting of fine beads to either camouflage or search a target object, enhance the local mixing of two different chemicals, and generate a desired concentrated chemical path. The magnetic composite elastomer is chosen here because it can be actuated and controlled wirelessly and fast by remote magnetic fields, which have minimal effects on the fluidic flow under investigation. Existing robots, including other jellyfish-like robots[1,2,5,30], could also complete the same tasks if they generate the same proposed lappet kinematics and local flow structures at the same length and time scale. Moreover, the proposed soft robotic platform, which has similar size and fluidic flow generating behaviors as an ephyra, could be used to study the impacts of changing their morphology and kinematics, which can happen due to pollutants, ionic changes, and temperature variation, to their survivability and habitat[24,31–34].

## Results

### Design and swimming behavior of the jellyfish-inspired swimming soft millirobot. 
*Scyphomedusae* ephyra (diameter: 1–10 mm) is characterized by its incomplete bell (Fig. 1a) and lappet paddling-based propulsion[23,25,26,28]. Inspired by such organism, our robot has a magnetic composite elastomer core (Supplementary Fig. 1), which can beat up and down its eight lappets in a non-reciprocal manner like an ephyra (Supplementary Fig. 4) under the control of an external magnetic field (**B**). As shown in Supplementary Fig. 2, each lappet has two compliant joints. Only the lappet distal joint is allowed to bend in the contraction phase, while both the distal and proximal joints of the lappet can bend in the recovery phase. This design induces a large wetted area during the contraction phase to acquire high thrust while decreasing the wetted area significantly during the recovery phase to reduce the drag force, just like the kinematics of a jellyfish ephyra (Supplementary Note 1). An air bubble of 0.3 μL is

introduced on top of the robot body by a pipette to reduce the robot's effective density to around 1.02 g·cm$^{-3}$.

With this soft robot design, we first design the external **B** to make the robot mimic the swimming mode of an ephyra studied by Feitl et al.[28] based on two common metrics quantifying the ephyra swimming kinematics: bell fineness and lappet velocity (Supplementary Note 5). The resulting biomimetic kinematics, referred as Mode A, are shown in Fig. 1b, c and Supplementary Movie 1. Keeping the beating frequency (2.5 Hz) and Reynolds number of the robot body ($Re_B = 7–95$) similar to an ephyra[25,27,28] (Supplementary Note 6), the robot can capture the typical flow structures of its biological counterpart[23,30]. As visualized by the particle image velocimetry (PIV) technique in the second row of Fig. 1b, the starting vortex forms at the beginning of the cycle (0 s) and dissipates quickly during the contraction (0–0.16 s). The stopping vortex forms a bit later than the starting vortex (0.08–0.16 s) and sheds during the recovery (0.24–0.32 s). This behavior contrasts to the well-formed starting-stopping vortex pair in an adult *scyphomedusae*[23,35,36]. During swimming, a portion of the surrounding water also propagates along with the robot due to the pressure field around the body[37], causing an upward drift flow below (indicated at 0.24 s). The flow structures are further visualized by a fluorescein dye, and the dye tree structure is observed to grow during swimming due to the induced drift and boundary layer shedding (the third row in Fig. 1b), similar to that created by an ephyra[29]. With such flow structures, the robot can trap objects from outside to the inside of the sub-umbrella region (Supplementary Fig. 3b) during propulsion (Fig. 1d), similar to the predation behavior of an ephyra[25].

### Five basic swimming modes and their propulsion performances. 
Apart from Mode A, we prescribe other four swimming modes (Modes B1, B2, B3, and C) with distinct fluidic flow generating behaviors and swimming performances by changing the lappet beating kinematics of the robot (Fig. 2; Supplementary Movie 2). First, we decrease the duration of the contraction phase ($t_C$) and recovery phase ($t_R$) to generate Mode B1 and Mode B2, respectively, while maintaining the lappet beating amplitude of Mode A. This change increases the angular velocity of the lappets in each mode ($\omega_C$ and $\omega_R$ are defined in the inset of Fig. 2b). As expected, the displacement per cycle increases when the robot beats faster in contraction and decreases when the robot beats faster in recovery (Fig. 2a). However, such difference does not reveal in the average velocity ($\bar{v}_{robot}$, mm·s$^{-1}$). In Fig. 2c, Mode B1 sees a significant increase in $\bar{v}_{robot}$ while $\bar{v}_{robot}$ for Mode B2 does not change significantly. This is because a shorter $t_R$ reduces the duration of a beating cycle ($t_C + t_R$) and makes the robot beat more frequently, compensating the loss in distance per cycle. Further reducing $t_R$, however, would eventually reduce $\bar{v}_{robot}$ to zero or even negative as the downward displacement in recovery would be equal to or even greater than the upward displacement in contraction. This trend can be explained by a dynamic model (Supplementary Figs. 10d and 10e and Supplementary Note 10). In addition to velocity, we also evaluate the propulsion efficiency by the reciprocal of the cost of transport (1/COT, Supplementary Note 7). As shown in Fig. 2c, Mode B1 results in a significant increase in 1/COT as the higher $Re_B$ associated with the higher $\bar{v}_{robot}$ benefits the robot more from the inertia. On the contrary, Mode B2 dispenses more energy during the recovery, which decelerates the robot, reducing the 1/COT.

Besides Mode B1 and Mode B2, the flexibility of this soft robot platform also allows combining swimming modes from other biological species into that of the ephyra. Therefore, we prescribe Mode B3 having an extra glide phase with duration $t_G$ after the contraction. Such a combination of stroke and glide phases is

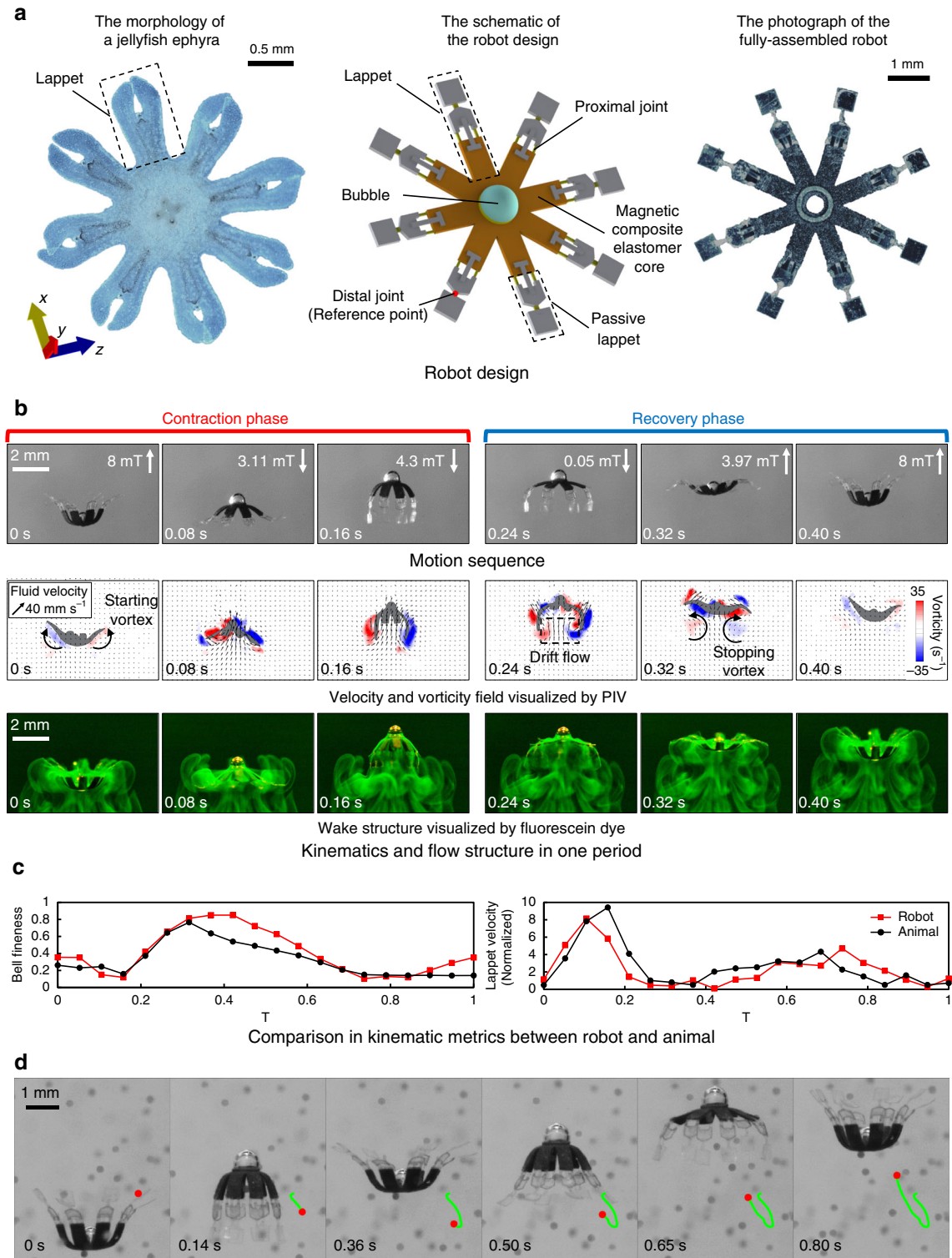

**Fig. 1** Design and swimming behavior of the jellyfish-inspired swimming soft millirobot. **a** Comparison in morphology. The newly budded *scyphomedusae* ephyra (*Aurelia aurita*) possesses deep clefts between two adjacent lappets. The design of the jellyfish-inspired soft millirobot captures such feature. The photo of the real animal is taken in a pet store and the animal's care is in accordance with the institutional guidelines. **b** Kinematics and flow structures achieved by Mode A. The motion sequence, the velocity and vorticity fields, and the wake structures visualized by the fluorescein dye are all in one cycle. The three experiments are from three different trials using robots with the same design and kinematics. **c** Comparison of the robot and animal in two kinematic metrics: bell fineness and lappet velocity. The biological data is reproduced with permission from Feitl et al.[28]; permission is conveyed through Copyright Clearance Center, Inc. **d** Video snapshots of capturing a neutrally buoyant bead using the fluid flow around the robot's lappets

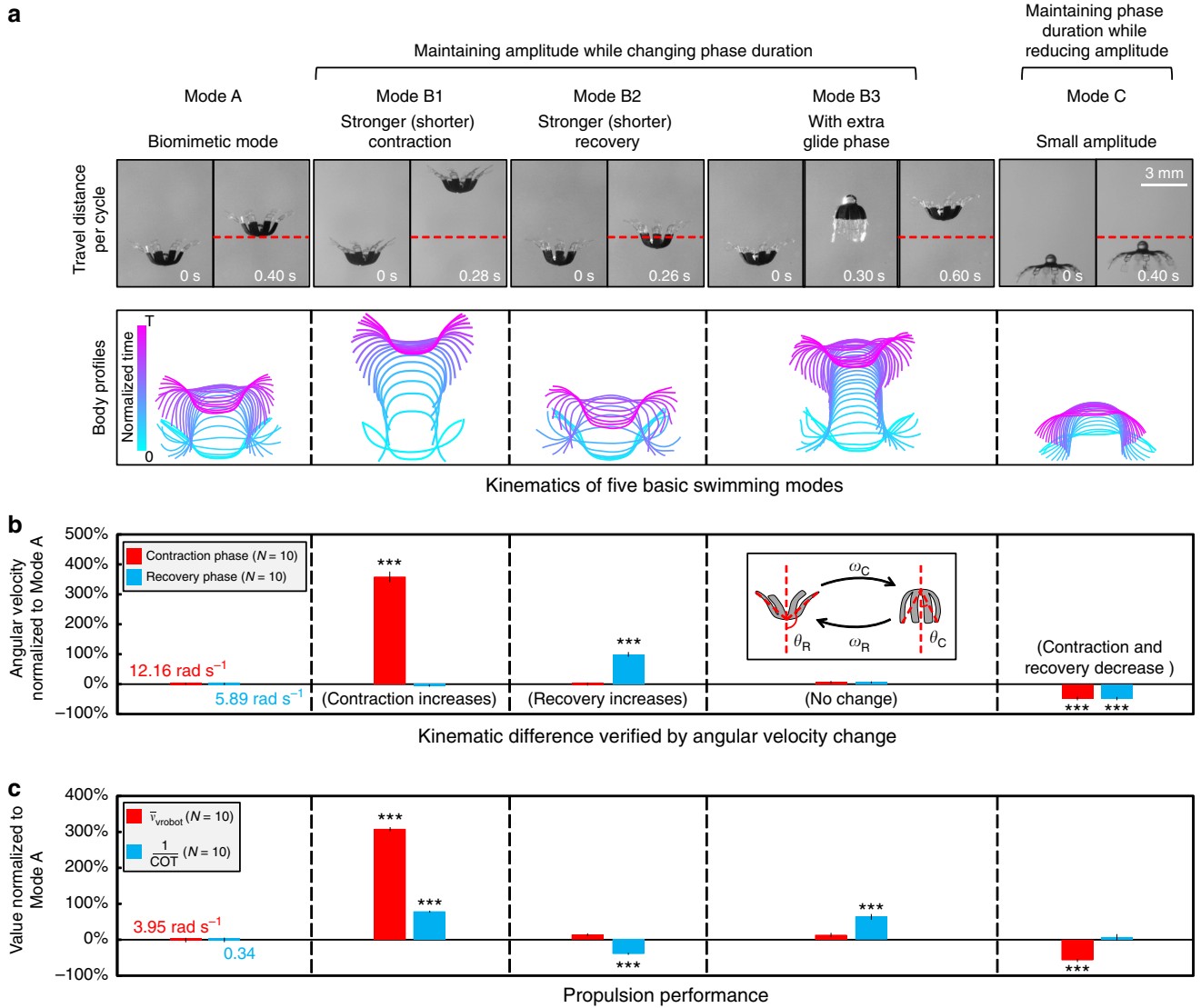

**Fig. 2** Five basic swimming modes being investigated and their impacts on swimming propulsion. **a** Kinematics of each swimming mode. The video snapshots of the start and end frames in one cycle are shown in the first row. One more frame is included for the glide phase of Mode B3. The red dashed lines indicate the final position of the robot in Mode A. The overlapped body profiles in one cycle are shown in the second row. **b** The difference in kinematics verified by angular velocity change. The beating angles and the angular velocities are defined in the inset. **c** The comparison in propulsion performance of each mode, showing the average velocity ($\overline{v}_{robot}$, mm·s$^{-1}$) and efficiency represented by the reciprocal of cost of transport (1/COT). In **b** and **c**, the error bars represent the standard error of the mean, and N is the number of trials

widely used by aquatic animals to save energy[38]. With such change, Mode B3 shows statistically significant increase in 1/COT (Fig. 2c), as the glide phase increases the displacement per cycle (Fig. 2a), while the work done within one cycle remains the same as Mode A. This result agrees with the fact that the inertia still plays an important role within the moderate $Re_B$[39]. However, $\overline{v}_{robot}$ does not increase significantly in Mode B3 as the peak speed, which is achieved at the end of the contraction, does not significantly change compared with Mode A. Moreover, further increasing $t_G$ would eventually slow down the robot due to the drag of the fluid and gravity (Supplementary Fig. 10f and Supplementary Note 10).

At last, we also prescribe Mode C with a smaller beating amplitude by decreasing the recovery angle $\theta_R$ while maintaining the contraction angle $\theta_C$ (defined in the inset of Fig. 2b), as previous literature shows that $\theta_R$ and the lappet beating amplitude ($\theta_R - \theta_C$) of the ephyra gradually decrease during its growth[27]. This mode can help us understand how the robot

performs with the kinematics of a larger-size ephyra than the one referred by Mode A[28]. Fig. 2c shows Mode C worsens $\overline{v}_{robot}$[23] while 1/COT does not have a significant change. According to the dynamic model, the beating amplitude of Mode A is very close to the optimal value that maximize the swimming velocity, and further increasing or decreasing the beating amplitude of Mode A both slow down the robot (Supplementary Fig. 10c and Supplementary Note 10).

**Object collection capability of the five basic swimming modes.** Changing the swimming kinematics of the robot also affects its object manipulation capability. First, the object collecting performance of the robot is quantified in Fig. 3 by the exchange rate of the water volume into the sub-umbrella region ($Q_{exchange}$, mm$^3$·s$^{-1}$). $Q_{exchange}$ is defined in Fig. 3b and implies how fast the water can be exchanged into the sub-umbrella region during one cycle, from the bottom boundary of the sub-umbrella. We assume

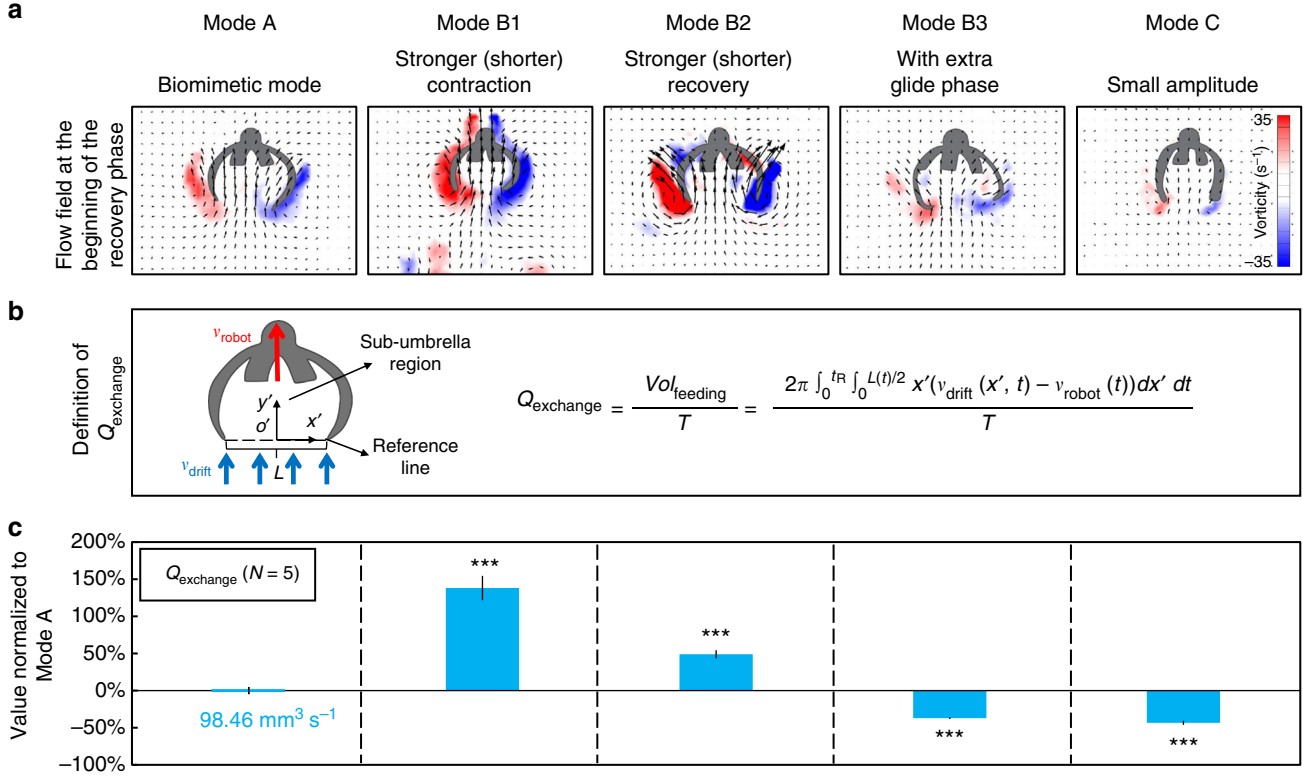

**Fig. 3** The object collecting performance of the five basic swimming modes. **a** The flow field at the beginning of the recovery phase. The feeding flow is integrated from this time instant until the end of the recovery phase to estimate the volume flow sucked in during the recovery. The stronger stopping vortex suggests a stronger feeding flow. **b** The definition of $Q_{exchange}$. The equation estimates the volume of the feeding flow $Vol_{feeding}$ exchanged into the sub-umbrella region by applying the axisymmetric assumption and integrating the 2D velocity obtained from PIV along the reference line. **c** $Q_{exchange}$ of the five basic swimming modes. A higher value indicates that the robot can collect objects faster. The error bars represent the standard error of the mean. $N$ is the number of trials

the same amount of water volume is sucked in during the recovery and expelled out during the contraction (Supplementary Note 9). Therefore, we only integrate the feeding flow ($v_{drift} - v_{robot}$) initiating from the beginning (Fig. 3a) to the end of the recovery. In nature, ephyra relies on this feeding flow to carry the preys into its sub-umbrella region for further capture and digestion[25].

As $Q_{exchange}$ is proportional to ($v_{drift} - v_{robot}$)/$T$, it can be increased by either increasing ($v_{drift} - v_{robot}$) to make the robot move slower to engulf faster upwards drifting flow, or by decreasing $T$ to increase the frequency of engulfing within a given time period. The experiment results support such prediction (Fig. 3c, Supplementary Fig. 9b). First, $Q_{exchange}$ increases significantly in both Mode B1, where $v_{drift}$ rises, and Mode B2, where $v_{robot}$ reduces. Second, the glide phase in Mode B3 has a negative contribution to $Q_{exchange}$ as an extra $t_G$ increases $T$ while ($v_{drift} - v_{robot}$) is similar to Mode A. Third, Mode C lowers $Q_{exchange}$ as $v_{drift}$ decreases and $v_{robot}$ increases during the recovery, which results in a smaller ($v_{drift} - v_{robot}$).

**Object retaining capability of the five basic swimming modes.** In addition to object collecting performance, Fig. 4 shows the object retaining performance of the robot investigated by tracing the trajectories of the trapped beads (Supplementary Movie 3). Neutrally buoyant beads are used here to exclude the effect of gravity. The results are quantified by the cycles and distance (mm) of a bead being retained by the robot. After being trapped inside the sub-umbrella region, a neutrally buoyant bead has two ways to escape (Fig. 4c). In Mechanism-1, the bead can escape with a probability, $P_C$, during the contraction phase. It acquires the momentum for escaping from the downward flow generated

by contraction and can be further enhanced by the beating of the lappets. In Mechanism-2, the bead can escape with a probability, $P_R$, during the recovery phase. It acquires the momentum for escaping from the stopping vortex and escapes from the distal tip of the lappet. Both $P_C$ and $P_R$ determine the retaining cycles and can be tuned with $t_C$, $t_R$ and $\theta_R$ ($\theta_C$ does not change in five basic modes) in our experiment. As accurately calculating $P_C$ and $P_R$ requires infinite trials of experiments, we can only obtain estimated $P'_C$ and $P'_R$ (Fig. 5g, see "Methods: Estimating the escaping probabilities"). As summarized in Figs. 5 and 6, weak contraction (larger $t_C$) and tighter sub-umbrella region (smaller $\theta_R$) make the robot retain the beads for more cycles while changing $t_R$ has contradicting effects on $P'_C$ and $P'_R$. A detailed discussion is provided below.

First, $P'_C$ and $P'_R$ increase with decreasing $t_C$ (Mode B1, Figs. 5a, g and 6a). When $t_C$ shortens, the downward fluidic flow during contraction becomes stronger and the chance of physical contact between the lappets and the beads increases, enhancing the Mechanism-1 and increasing $P'_C$. In addition, the strong stopping vortex (Supplementary Fig. 8) induced by high propulsion speed circulates more beads out of the sub-umbrella region during the upcoming recovery phase, enhancing the Mechanism-2 and increasing $P'_R$. With both $P'_C$ and $P'_R$ increase, the object retaining cycles decrease (Fig. 4b). However, the overall retaining distance of Mode B1 does not decrease significantly, as Mode B1 makes the robot have a longer displacement per cycle (Fig. 2a), compensating the loss in the retaining cycles.

Second, $P'_R$ increases with decreasing $t_R$ (Mode B2, Figs. 5b, 5g, and 6b) as shorter $t_R$ creates a stronger stopping vortex (Supplementary Fig. 8) and more beads could be circulated out

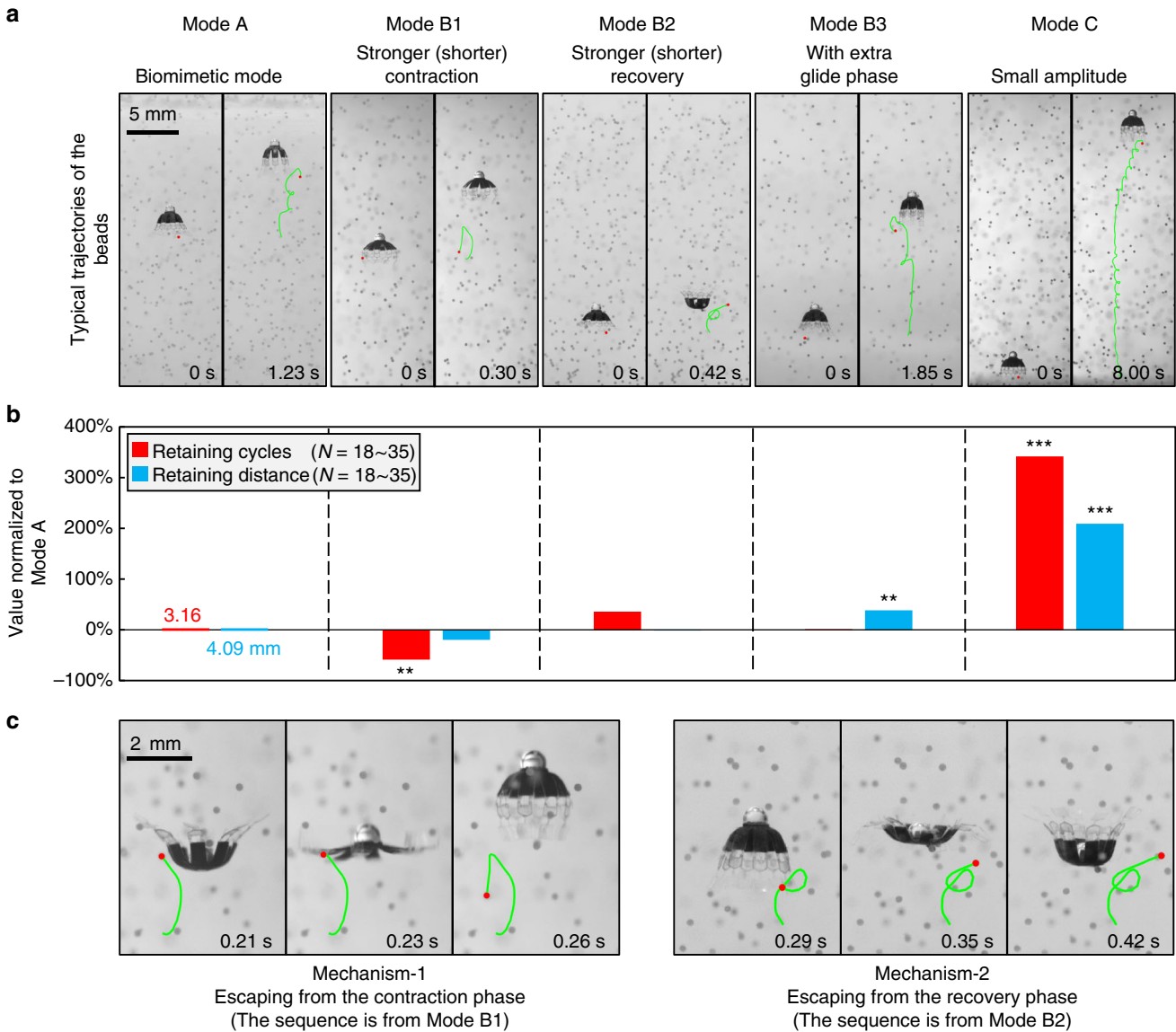

**Fig. 4** The object retaining performance of the five basic swimming modes. **a** The typical translation trajectories of neutrally buoyant beads. In the left frame, the bead just enters the sub-umbrella region. In the right frame, the bead completely escapes the control of the robot. **b** The retaining metrics quantified by neutrally buoyant beads. N is the number of the counted beads. **c** Two escaping mechanisms. The trajectory sequence of Mechanism-1 is from Mode B1. The trajectory sequence of Mechanism-2 is from Mode B2. The two sequences are chosen as each mechanism is most typical in each corresponding mode

from the distal tip of the lappet during recovery through Mechanism-2 (Fig. 5f), increasing $P'_R$. In contrast, $P'_C$ decreases with decreasing $t_R$ as reducing $t_R$ increases the recapture of the beads beaten out during the last contraction phase, weakening the Mechanism-1 (Fig. 5e, see "Methods: Bead trajectory tracing experiments"). Therefore, in Mode B2, the increase in $P'_R$ compensates the loss in $P'_C$. Consequently, the retaining cycles do not show statistically significant change (Fig. 4b).

Third, $P'_C$ and $P'_R$ decrease with decreasing $\theta_R$ (Mode C, Figs. 5d, g and 6d). Reducing $\theta_R$ solely decreases $\omega_C$ and $\omega_R$, which consequently reduces the magnitude and scale of the vortices incurred (Supplementary Fig. 8). Therefore, the beads circulate slower inside the sub-umbrella region and are less likely to escape through both Mechanism-1 and Mechanism-2. Moreover, a smaller $\theta_R$ creates a tighter sub-umbrella region during contraction, and it squeezes the beads towards the robot central axis, where a stronger upward drifting flow exists (Fig. 1b). This behavior increases the chance of the beads being recaptured by

the upcoming recovery phase, decreasing $P'_C$ (Fig. 5e, see "Methods: Bead trajectory tracing experiments"). During the recovery, the tighter sub-umbrella region makes the trapped beads have more chances to collide with the inner wall of the sub-umbrella region when they try to escape through the Mechanism-2, decreasing $P'_R$. With $P'_C$ and $P'_R$ both reduced, Mode C greatly increases the retaining cycles and distance of the trapped beads (Fig. 4b).

At last, increasing $t_G$ does not alter $P'_C$, $P'_R$ (Mode B3, Figs. 5c, g and 6c) and hence the retaining cycles (Fig. 4b). There is no fluid exchange during gliding (Supplementary Fig. 9b) and consequently, the beads can keep their positions relative to the robot. Therefore, Mode B3 is observed to have no significant change in retaining cycles in comparison with Mode A. For each retaining cycle, however, the beads can travel a longer distance as it is possible to tune $t_G$ to make the robot have longer displacement per cycle than Mode A (Supplementary Fig. 10f), which is the case for Mode B3 (Fig. 2a).

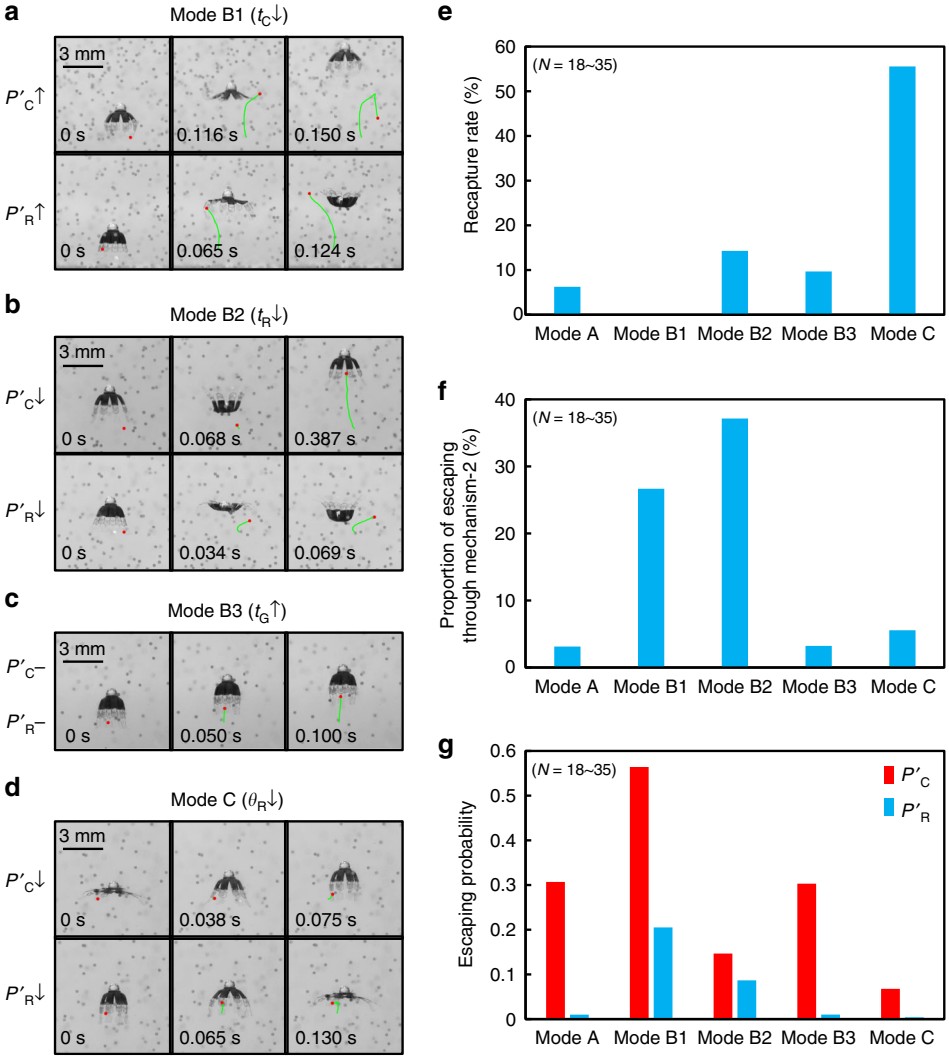

**Fig. 5** The influence of changing the kinematic parameters on object manipulation capability. **a–d** Typical trajectories of the neutrally buoyant beads achieved by different swimming modes. The changing in escaping probabilities are marked on the left. **e** Recapture rate of beads being transported. A higher value indicates a less chance to escape through Mechanism-1. **f** Proportion of the beads escaping through Mechanism-2. A higher value indicates a higher chance to escape through Mechanism-2. **g** The escaping probabilities estimated from experiments. In **e–g**, $N$ represents the number of the counted beads

Note that the probabilistic estimate of the bead transportation is based on idealized assumptions. More experiments are needed to find the extent to which the $P'_C$ and $P'_R$ are still valid, e.g., in more complex conditions.

**Four robotic tasks realized by newly prescribed modes**. The swimming propulsion and object manipulation performances of the above five basic swimming modes are summarized in Table 1. Table 1 shows that no swimming mode performs best in all aspects. For example, Mode B1, who is better in propulsion and object collecting while worse in object retaining, contrasts with Mode C, who is worse in propulsion and object collecting while better in object retaining. Based on Table 1, we prescribe additional new modes (Modes D, E, F, and G) with different lappet kinematics to enable the robot to achieve specific tasks, other than swimming, as shown in Fig. 7.

As the first task, the robot can selectively transport beads with different sizes from the bottom to the top of the water tank using the proposed Mode D1 (trapping large beads while expelling small beads) and Mode D2 (trapping small beads while leaving large beads) in Fig. 7a (Supplementary Movie 4). Such selective transportation is useful to transport and deliver drug type of cargos[6] in biomedical applications, collect samples with specific size[1,40] for environmental monitoring, and clean microplastics for environmental cleaning[41]. Beads heavier than water (density: $1.05\,\mathrm{g\cdot cc^{-1}}$) are used here to ensure that they initially stay at the tank bottom for fair comparison. For such beads, gravity enables a third escape mechanism (Mechanism-3) if the beads carried by drift flow cannot catch up with the moving robot. Through an analysis of the drag force and gravity, we know that when the densities are the same, the larger beads are easier to escape through the Mechanism-3, and hence requires a faster average feeding flow speed ($\bar{v}_{feeding}$) to catch up with the robot. A thorough discussion on the mechanism of selective transportation and prescribing kinematics can be found in Supplementary Note 12. In summary, we prescribe Mode D1 with a larger $\bar{v}_{feeding}$ to trap and transport the large beads. While the small beads are expelled due to the shorter relaxation time. In contrast, by reducing $\theta_R$ relative to Mode D1, we prescribe Mode D2 with smaller $\bar{v}_{feeding}$

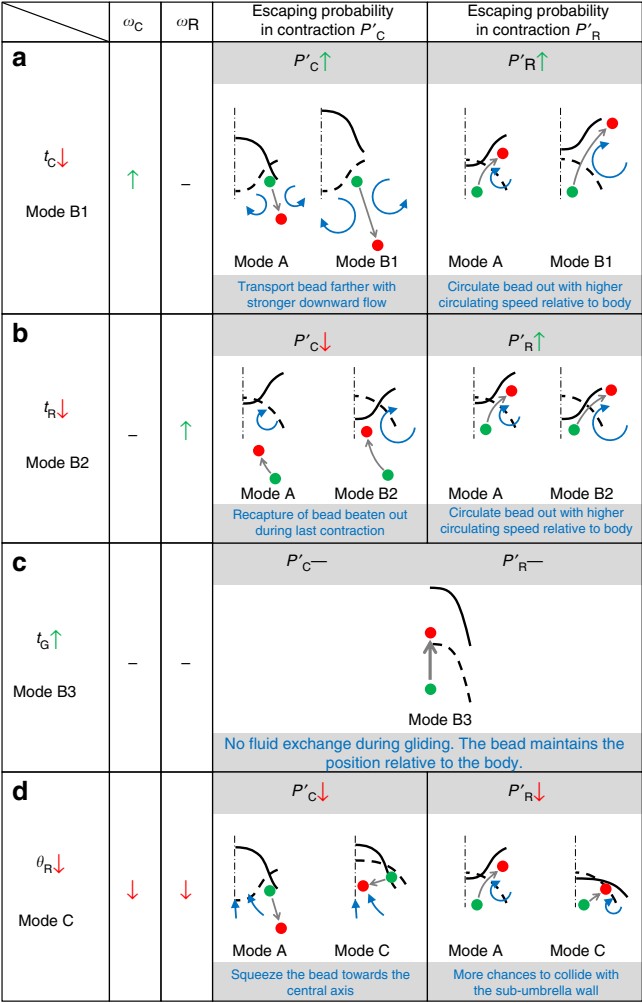

**Fig. 6** The mechanisms of changing the escaping probability by changing the robot's kinematic parameters. The comparisons between Modes B1, B2, B3, and C with Mode A are shown in **a-d**, respectively. The green up arrow, the red down arrow, and the black line indicate, respectively, the increase, decrease, and no change in kinematic parameter value or probability. The dashed and solid curves indicate, respectively, the positions of the robot body at the last and the current phases. The green and red dots indicate, respectively, the positions of the object at the last and the current phases

and better object retaining capability. Therefore, the robot can leave behind the large beads while retaining small beads. Such selective trapping and transportation is repeatable and quantified in Supplementary Figs. 18 and 20.

As the second task, the robot can also burrow, by interacting with the solid granular medium on the bottom surface. Inspired by many sand-dwelling animals that burrow to predate or to escape from the predators[42,43], here we show that our robot can burrow into granular media to either camouflage or search a buried object under the fine beads using Mode E (Fig. 7b, Supplementary Movie 5). To realize burrowing, the recovery phase of Mode E is prescribed to be much stronger than the contraction phase ($\omega_C = 17.36\ \mathrm{rad \cdot s^{-1}} < \omega_R = 39.31\ \mathrm{rad \cdot s^{-1}}$) and to reach the largest $\theta_R$ that can be achieved in our system ($\theta_R = 2.63$ rad). Therefore, the robot cannot propel upwards and stays at the bottom. The stronger recovery than any other swimming modes also makes the beads under the body easier to be expelled through the Mechanism-2. Some of the beads are expelled directly to the side of the robot (Supplementary Fig. 21b). Other beads are first expelled up and

then beaten away further by the lappets during the following recovery phases (Supplementary Fig. 21c). For camouflage, the robot is required to bury itself. Therefore, the robot is positioned flat. The beads under the robot are first expelled up, and then they gradually settle down and bury the robot body. For object searching, however, the robot is required to expel out the beads with a tilted angle to avoid the beads falling back and covering the target object again. Therefore, the robot is tilted to expel beads obliquely. The robot eventually finds the target black bead and expels it out. A detailed discussion on the burrowing process can be found in Supplementary Note 13.

As the third task, we demonstrate that the robot can enhance the local mixing of the fluids using Mode F (Fig. 7c, Supplementary Movie 6), which is inspired by the discovery that the swimming of the ephyra helps to enhance the mixing of the ocean at the moderate $Re^{29}$. Many marine species, such as asteroids, sea urchin and some corals, reproduce externally by releasing gametes into the surrounding flow[44–46], and the successful fertilization relies on the sperm-egg contacts[45]. Finding effective kinematics for untethered miniature swimming robots to mix the fluids locally may potentially help these organisms to increase the contact chance of the gametes, boosting their reproduction. Compared with Mode A, Mode F increases both the contraction ($\omega_C = 27.61\ \mathrm{rad \cdot s^{-1}}$) and the recovery ($\omega_R = 27.70\ \mathrm{rad \cdot s^{-1}}$) to locally enhance mixing. With the recovery phase being slightly stronger than the contraction phase, the robot can suspend at the bottom center of the tank. During mixing, the robot first draws dyes from both sides during recovery, then squeezes the dyes from both sides to the sub-umbrella region by contraction, and finally redistributes the mixed dye back to the environment through Mechanism-1 and Mechanism-2 (Supplementary Fig. 22). A detailed discussion on the mixing process can be found in Supplementary Note 14.

As the last task, we demonstrate that the robot can generate a desired chemical path in its wake by using Mode C (Fig. 7d, Supplementary Movie 7). This function could be useful in spreading pheromones (or other specific chemicals) into desired positions, by which the robot could intentionally interact with aquatic animals by controlling their migration, mating, and various social behaviors[47]. We choose Mode C to finish this task as it has the best object retaining performance (Table 1) and can better resist the spreading of the chemicals. When swimming upwards with Mode C from the dye bolus injected on the tank bottom, the robot can create a straight and concentrated chemical path in comparison with other basic modes (Supplementary Fig. 23b). We then show that the chemical path can also be generated as a more complex S-shaped trajectory by using the external magnetic field to steer the robot in two dimensions by Mode G (Fig. 7d). In comparison with Mode C, Mode G lets the robot beat and swim faster with a smaller sub-umbrella region. A detailed discussion on creating the chemical path can be found in Supplementary Note 15.

## Discussions

The untethered jellyfish-like soft millirobot, which has similar size and fluidic flow generation behavior as an ephyra, can achieve diverse physical functions and robotic tasks by manipulating its surrounding fluidic flow. The ability to utilize the fluidic flow to achieve multiple functions and tasks is independent of the magnetic field since the incurred flow structures only rely on the interplay between the robot body and the fluid. Therefore, the above design and swimming modes may potentially be realized by current or future jellyfish-like soft robots built with other on-board or off-board actuation methods[48] such as biological muscle cells[30], shape memory alloys[5], hydraulic

**Table 1 Summary of propulsion and object manipulation performances of different basic swimming modes**

| Mode | Kinematic parameters being changed | Angular velocity | | Propulsion performance | | Object manipulation performance | | |
| --- | --- | --- | --- | --- | --- | --- | --- | --- |
| | | | | | | Object collecting | Object retaining | |
| | | $\omega_R$ | $\omega_C$ | $\overline{V}_{robot}$ | $\frac{1}{COT}$ | $Q_{exchange}$ | Retaining cycles | Retaining distance |
| B1 | $t_C \downarrow$ | — | ↑ | ↑ | ↑ | ↑ | ↓ | — |
| B2 | $t_R \downarrow$ | ↑ | — | — | ↓ | ↑ | ↓ | — |
| B3 | $t_G \uparrow$ | — | — | — | ↑ | ↓ | — | ↑ |
| C | $\theta_R \downarrow$ | ↓ | ↓ | ↓ | — | ↓ | ↑ | ↑ |

*Note*: The increase and decrease in kinematic parameter values or angular velocities are, respectively, marked by black up and down arrows. The increase and decrease in performance metrics are, respectively, marked by green up and down arrows. The black lines indicate no statistical difference in kinematic parameter values, angular velocities, or performance metrics

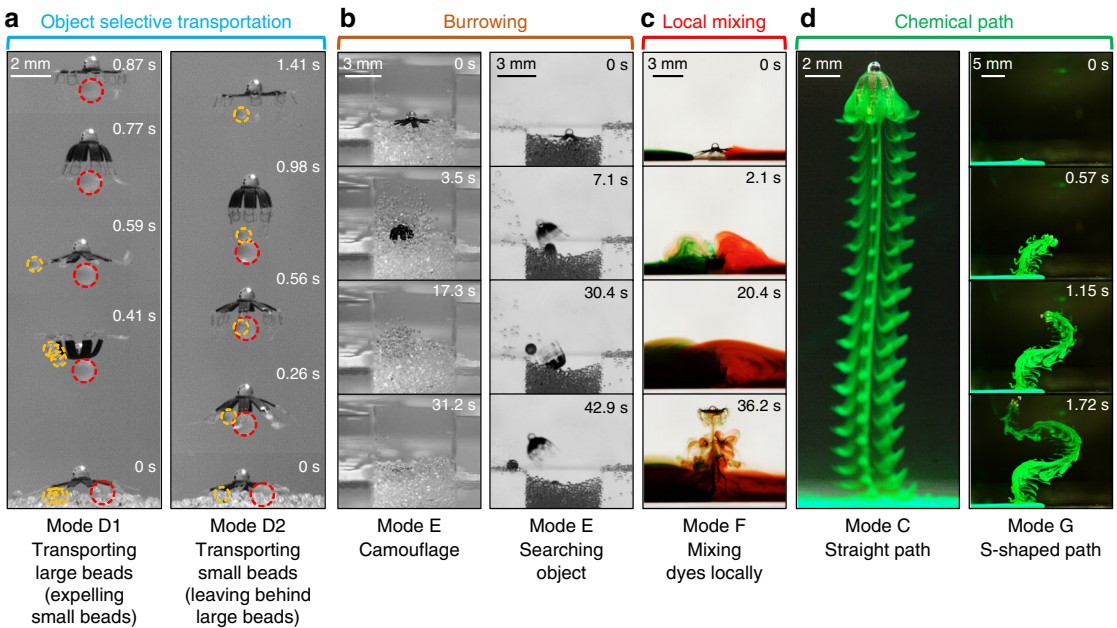

**Fig. 7** Four tasks realized by newly prescribed modes extended from the five basic modes. **a** Object selective transportation. The robot can use Mode D1 to transport large beads (diameter: 0.99 mm ± 0.025 mm, density: 1.05 g·cc⁻¹) while expelling small beads (diameter: 500–600 μm, density: 1.05 g·cc⁻¹). It can also use Mode D2 to transport small beads and leave behind large beads. **b** Burrowing for camouflage and searching in fine beads (diameter: 200–300 μm density: 1.05 g·cc⁻¹) with Mode E. **c** Locally mixing two food dyes with different colors using Mode F. **d** Generating a desired concentrated chemical path. The robot can swim upwards to create a straight chemical path with Mode C. It can also be steered in 2D to create a desired S-shaped chemical path with Mode G

actuators[1], dielectric elastomers[2,49], hydrogels[50], and liquid crystal elastomers[51].

In addition, this robotic platform could also be used as a scientific tool to further study the behaviors of ephyrae due to its various advantages, such as the ability to change the locomotion mode on demand and not being influenced by physiological factors[52,53]. Ephyrae hold a critical position in ocean ecological system due to their large quantity and wide distribution[54]. Although many previous researches have studied ephyrae's most typical swimming kinematics[23,25–29], the impacts of changing the morphology and kinematics of the ephyrae to their survivability and habitat, which can happen when the environment is affected by pollutants[31–33], ion concentration change[34], and temperature variation[24], remains to be further investigated using such tool as a future work.

## Methods

**Magnetic composite elastomer core**. The detailed design of the magnetic composite elastomer core is shown in Supplementary Fig. 1a. The core has a thickness

of 65 μm. The circumscribed circle of the core has a diameter of 3 mm. A circular hole with a diameter of 0.5 mm is designed in the robot center to trap the bubble. To ensure that the bubble trapped on the top does not go through the hole to the bottom, an elastic ring with a thickness of 65 μm is designed to be on the top center of the magnetic composite elastomer core.

The magnetic composite elastomer has been reported in our previous work[7]. It is a composite of the soft silicone rubber (Ecoflex 00-10, Smooth-On Inc.) and the neodymium-iron-boron (NdFeB) magnetic microparticles (MQP-15-7, Magnequench; average diameter: 5 μm) with a mass ratio of 1:1. The resulting magnetic elastomer has a density of 1.86 g·cm⁻³. The mixture is then cast onto a flat poly (methyl methacrylate) plate coated with a thin layer of parylene C (6 μm thick) to form a thin film with a thickness of 65 μm. The material is then put into the oven under 60 °C for curing around 1 h. After curing, the magnetic part is cut out by using the UV laser. The elastic ring for fixing the bubble position is nonmagnetic and is made of Ecoflex 00–10 loaded with aluminum powder (5413 H Super, laborladen.de) in a weight ratio of 1:2. It is fabricated in a similar way as the magnetic core and then glued to the magnetic composite elastomer core by Ecoflex 00–10.

The magnetic composite elastomers mentioned above are hydrophobic, which is determined through the sessile droplet method. The static water contact angle of the material used to build the magnetic composite elastomer core is characterized to be 108 ± 3°. The static water contact angle of the material used to build the elastic ring is characterized to be 110 ± 5°. To create the magnetization profile of

the core, a water droplet is pipetted on the core (Supplementary Fig. 1c). The core can automatically wrap the droplet due to the attraction of the water droplet[55], forming an ellipsoidal shape (Supplementary Figs. 1d and 1e). The volume of the water droplet is well controlled to be 1 μL using a pipette. We then put the core along with the water droplet in a freezer until the water droplet is totally frozen. This is to fix the ellipsoidal shape of the core during the magnetization process. We finally apply a strong uniform **B** field (1.8 T) inside a vibrating sample magnetometer (VSM, EZ7, Microsense) in the direction shown in Supplementary Fig. 1d, which makes the magnetization magnitude to be $71{,}700 \pm 1725$ A·m$^{-1}$ and generates the magnetization profile shown in Supplementary Fig. 1f. With such magnetization profile, the magnetic composite elastomer core can deform upwards when $B_y > 0$, and deform downwards when $B_y < 0$ (Supplementary Fig. 1b). When $B_y = 0$, however, the magnetic composite elastomer core still shows a curvature. As discussed in our previous work[7], such deformation at the rest state may be caused by the residual strain energy due to the fabrication process. We currently have not observed any influence of this residual strain energy on our experiment results.

The robot can be steered by rotating the direction of the applied **B** field. With an additional horizontal pair of coil set, we can steer the robot in the 2D plane by applying magnetic torque (Supplementary Movies 1 and 7). In the last scene of Supplementary Movie 1, the robot toddles slightly when it tilts too much from swimming vertically. However, this does not affect the steering of the robot. There are two possible reasons for this. First, the deformation of the magnetic composite elastomer core is not perfectly axial symmetric, causing the net magnetic moment deviating from its central axis and inducing the undesired magnetic torque. Second, this might be due to the bubble, which is trapped on top of the body, providing a self-righting torque, which always rotates the robot central axis back to be vertical. We observe that such phenomenon relieves when the beating frequency increases and beating amplitude reduces as in Supplementary Movie 7. Such dynamic process would be further investigated in the future.

**Passive lappets**. The detailed design of the passive lappets is shown in Supplementary Fig. 2a. Each passive lappet is composed of five parts: T stopper, proximal pad, distal pad, proximal joint and distal joint. The T stopper, proximal pad, and distal pad are all made of parylene C. They are cut out from a layer of parylene C with a thickness of 6 μm. The proximal and distal joints linking the proximal and the distal pads are made of Ecoflex 00–10. They have an average thickness of 15 μm. The T stopper can restrict the upward bending of the proximal joint while having little influence on the downward bending. With such mechanism, the proximal joint bends less during the contraction phase than during the recovery phase (Supplementary Fig. 2c).

With the robot center pinned on a pillar (restricting only the vertical translation), the frequency response of the whole lappet, including both magnetic lappet and passive lappet, is tested under sinusoidal $B_y$ fields of three different magnitudes (10, 20, and 30 mT) and different frequencies (0.5–30 Hz) (see Supplementary Fig. 2b). Five measurements are conducted for each case. In this report, we borrow the concept of cut-off frequency from the linear system to clarify the relation between the actuating magnetic field (magnitude and frequency) and the resultant beating amplitude. We define the cut-off frequency of the lappet to be the frequency at which the beating amplitude, $\theta_R - \theta_C$, is 0.707 times that achieved under 0.5 Hz. Linear least square regression is used to obtain the relation between the actuation frequency and the beating amplitude to find out the cut-off frequency (Supplementary Fig. 2b). In Supplementary Fig. 2b, we show that the cut-off frequency of the lappet can be increased by increasing the **B** field magnitude. In the future, the non-linearity in frequency response (e.g., the frequency response observed at 10 mT) will be investigated.

**Electromagnetic coil setup and particle image velocimetry system**. The schematic of the experimental setup for quantitative characterization experiments is shown in Supplementary Fig. 3a. The two electromagnetic coils providing the uniform vertical external magnetic field are arranged in the Helmholtz configuration. 95% homogeneous region of the coils is measured to be 45 mm along $Y$ direction. The largest |**B**| it can provide is 30 mT. During the experiment, the robot swims in a transparent water tank situating at the central region of the coil system. The dimension of the tank used for quantifying the five basic swimming modes is $100 \times 60 \times 40$ mm$^3$ (length × width × height). To minimize the viscosity change due to the temperature variation and guarantee the comparability of the results, the coil system is water cooled, and all the experiments are conducted at around 23 °C. To minimize the wall and surface effects, only the experimental data obtained when the robot swims at least 5 mm away from the bottom and the water-air interface are adopted. The fluid flow around the robot is characterized by a PIV system (Dantec Dynamics, Inc.). The water is evenly seeded with 1 μm-diameter polystyrene particles loaded with fluorochrome dye (Molecular Probes, Inc., Eugene, OR, USA; $1.1 \pm 0.035$ μm) which can be excited with a laser at 535 nm wavelength and emits fluorescence at 575 nm. The laser beam (1000 Hz, 527 nm) is expanded into a plane and projected vertically from the water tank bottom. The movement of the particles is captured using a high-speed camera (M310, Phantom, Inc.). A 570 nm high-pass lens filter is used to increase the contrast between the PIV particles and the background. The image sequences recorded are then processed in the commercial software (DynamicStudio 2016a, Dantec Dynamics, Inc.) to obtain velocity fields by applying a cross-correlation algorithm. The characterization experiments are all

performed in this setup using multiple robots with the same design. No fatigue of the robot has been observed throughout our experiments.

Currently, the control of the robot is realized by providing an oscillating magnetic field along its body central axis. Therefore, the controllable degree of freedoms (DOFs) of the robot's rigid-body translational and rotational motions depend on the configuration of the electromagnetic coil system. With a single pair of fixed electromagnets, we have one control DOF for controlling the 1-DOF rigid body translational motion (e.g., swimming vertically in characterization experiments). With one more pair of fixed electromagnets, one more controllable DOF for rotational motion can be obtained and steering the robot in 2D can be achieved (e.g., generating S-shaped chemical path). To improve the control performance, we can add more coils to the system[56,57], improve the dynamic model (Supplementary Note 10), and implement visual feedback control.

**Definition of the sub-umbrella region**. The sub-umbrella or bell region is the area used by the robot to trap objects, similar to ephyra trapping its prey. The sub-umbrella region is defined in Supplementary Fig. 3b. When the robot deforms its body into a bell shape (0.08–0.24 s in Supplementary Fig. 3b), the sub-umbrella region is defined to be the enclosed region between a reference line that links the lappet tips of both sides and the whole robot body[30]. When the robot deforms into an inversed-bell shape (0 s, 0.32–0.40 s in Supplementary Fig. 3b), the sub-umbrella region is defined to be the area enclosed by the robot body and a horizontal reference line that is 300 μm away from the bottom of the robot. 300 μm distance is selected here to accommodate at least one small bead (used in bead trajectory tracing experiments, 212–250 μm in diameter) right below the robot's body.

**Kinematics of five basic swimming modes**. We use two methods to tune the five basic swimming modes. All the control signals used are shown in Supplementary Fig. 5d. In the first approach, we tune the duration of each phase of Mode A while maintaining the beating amplitude to generate other three basic swimming modes (Modes B1, B2, and B3). Specifically, Mode B1 has a shorter contraction ($t_C$), 21.76% of that in Mode A, while the recovery duration ($t_R$) is kept unchanged. Consequently, the average angular velocity of the contraction $\omega_C = (\theta_R - \theta_C) / t_C = 55.85$ rad·s$^{-1}$ is larger than $\omega_C = 12.16$ rad·s$^{-1}$ of Mode A, and the contraction is more powerful. Mode B2 has a shorter recovery ($t_R$), and its recovery phase is 50.13% of that in Mode A, while the $t_C$ is the same. Consequently, the angular velocity of the recovery $\omega_R = (\theta_R - \theta_C)/t_R = 11.69$ rad·s$^{-1}$ is larger than $\omega_R = 5.89$ rad·s$^{-1}$ of Mode A, and the recovery is more powerful. Mode B3 has an extra glide phase compared with Mode A, and the glide duration $t_G = 0.2$ s.

In the second approach, we prescribe Mode C with smaller beating amplitude by decreasing $\theta_R$ while maintaining $\theta_C$ and the duration of each phase the same compared with Mode A. Consequently, the average velocities of both the contraction and the recovery phases ($\omega_C = 6.27$ rad·s$^{-1}$, $\omega_R = 3.04$ rad·s$^{-1}$) are weaker than Mode A.

We do not change the lappet kinematics by simply changing the actuation frequency, $f$, because of the following reasons. First, since the beating period $T = 1/f = t_C + t_R + t_G$, changing the actuation frequency $f$ alters $t_C$, $t_R$ and $t_G$ at the same time. Second, only changing $f$ can also change the beating angles $\theta_R$ and $\theta_C$, as well as the beating amplitude $\theta_R - \theta_C$ (Supplementary Fig. 2b). Therefore, only changing $f$ is not appropriate if the impacts of each kinematic parameter ($t_C$, $t_R$, $t_G$, $\theta_R$, and $\theta_C$) are to be investigated. Also, see Supplementary Notes 3 and 4 for the details of tuning individual kinematic parameters while maintaining others.

Note that $\theta_C$ doesn't change for all characterization modes. Under the current experiment setup, the minimum $\theta_C$, the maximum $\theta_R$, the maximum $\omega_C$, and the maximum $\omega_R$ that can be achieved are, respectively, 0.44 rad, 2.63 rad, 55.85 rad·s$^{-1}$, and 39.31 rad·s$^{-1}$.

**Bead trajectory tracing experiments**. The bead trajectory tracing experiments shown in Fig. 4 are conducted in a transparent water tank with a size of $100 \times 60 \times 40$ mm$^3$ (length × width × height). Polystyrene beads with the diameter of 212–250 μm and density of 1.00 g·cc$^{-1}$ (Cospheric, Inc.) are used to exclude the effect of the gravity. This is in accordance with the natural preys of ephyrae, as their sizes range from 100 to 5000 μm[26] and are often regarded as neutrally buoyant[58]. In each experiment trial, the robot swims upwards from the tank bottom, and the beads scattered in the water are randomly captured by the robot. The motion of the robot and the beads are captured by a high-speed camera with a frame rate of 500 frames per second (fps). For each swimming mode, the experiment is repeated for 5–10 times. See Fig. 4a, Fig. 5a-d, and Supplementary Movie 3 for the typical transportation trajectory of the bead in each mode.

We manually trace the trajectories of the beads captured into the sub-umbrella region and use two metrics, retaining cycles and retaining distance (mm), to quantify the object retaining performance. The retaining cycle is defined as the cycle number of a bead being retained during the trapping process. The retaining distance is defined as the distance of a bead being transported during the trapping process. The trapping process begins at the time instant when the bead is first captured into the sub-umbrella region and terminates when the beads completely escape. Here, the word 'completely' means the escaped beads will not be recaptured into the sub-umbrella region during the rest of the trial (the end of the whole upwards swimming process).

Apart from the metrics defined above, we also quantify two indicators: the recapture rate and the proportion of escaping through Mechanism-2 (Fig. 5e, f). Recapture usually happens to a bead that is beaten out during the contraction (Mechanism-1) but is still near the drift flow of the robot. The upcoming recovery phases can then pull the bead back into the sub-umbrella region. The recapture rate indicates the proportion of beads that are recaptured during the transportation process, among all the beads that are transported. It can be used to indicate whether the escaping through Mechanism-1 is reduced. The proportion of escaping through Mechanism-2 evaluates the proportion of the beads that escape through Mechanism-2, among all the beads that escape. A higher value indicates that a trapped bead has a higher chance to escape during the recovery.

Since the beads are captured randomly, the number of the counted beads in different swimming modes are different. In Modes A, B1, B2, B3, and C, the number of the counted beads are respectively 32, 30, 35, 31, and 18.

**Estimating the escaping probabilities**. The process of the bead transportation shown in our experiments (Fig. 4) is very complicated. The neutrally buoyant beads are randomly seeded in the water and are randomly trapped by the robot. It is hard to find a simple and deterministic rule to predict the trajectory of each bead being trapped. Since we know little about the whole transporting process, probabilistic models are applicable to describe the complex behavior of the system[59]. As for a neutrally buoyant bead, we assume each swimming cycle of the robot is an independent event that can produce two outcomes. In the first outcome, the trapped bead escapes the sub-umbrella region with probability $P_{out}$. In the second outcome, the trapped bead is still retained within the sub-umbrella region with probability $1 - P_{out}$. If we further assume the probabilities of these two outcomes hold constant, the number of the beating cycles needed to expel a trapped bead has a geometric distribution. The expected number of the retaining cycles can then be expressed as:

$$E_{retain} = \lim_{k \to \infty} \sum_{n=1}^{k} n(1 - P_{out})^{n-1} P_{out} \tag{1}$$

In the first outcome, the escape probability $P_{out}$ can be expressed as $P_{out} = P_C + P_R$ because the beads can escape either through Mechanism-1 with probability $P_C$ (during contraction) or through Mechanism-2 with probability $P_R$ (during recovery). With the above assumptions, Equation 1 can be formulated as:

$$
\begin{aligned}
E_{retain} &= \lim_{k \to \infty} \sum_{n=1}^{k} (1 - P_{out})^{n-1} (P_C + P_R) \\
&= P_C \lim_{k \to \infty} \sum_{n=1}^{k} (1 - P_{out})^{n-1} + P_R \lim_{k \to \infty} \sum_{n=1}^{k} (1 - P_{out})^{n-1} \\
&= E_{retain-1} + E_{retain-2}
\end{aligned}
\tag{2}
$$

where $E_{retain-1}$ and $E_{retain-2}$ are, respectively, the expected number of the retaining cycles of a bead that is expelled through Mechanism-1 and Mechanism-2. $E_{retain-1}$ and $E_{retain-2}$ can further be derived as:

$$
\begin{aligned}
E_{retain-1} &= \lim_{k \to \infty} P_C \left( \frac{1 - (1 - P_{out})^k}{P_{out}^2} - \frac{k(1 - P_{out})^k}{P_{out}} \right) \\
E_{retain-2} &= \lim_{k \to \infty} P_R \left( \frac{1 - (1 - P_{out})^k}{P_{out}^2} - \frac{k(1 - P_{out})^k}{P_{out}} \right)
\end{aligned}
\tag{3}
$$

Because $0 < P_{out} < 1$ and $k \to \infty$, we can obtain $E_{retain-1} = \frac{P_C}{P_{out}^2}$ and $E_{retain-2} = \frac{P_R}{P_{out}^2}$ from Eq. 3. As it is impossible to implement infinite trials of experiments ($k \to \infty$), theoretically we cannot get $P_C$ and $P_R$ from the experiment results. However, if we assume the average retaining cycles of the trapped beads from the beads tracing experiments (Fig. 4b) as reasonable estimations to $E_{retain-1}$ and $E_{retain-2}$, then we can obtain probabilities $P'_C$ and $P'_R$ for each swimming mode (Fig. 5g). Therefore, if $P'_C = P_C$ and $P'_R = P_R$, then $E_{retain-1}$ and $E_{retain-II}$ are the expected value for the cycling number recorded in the experiment.

In fact, the last beating cycle does have an influence on the following beating cycle, since the initial position and velocity of the bead of the following cycle are influenced by the last beating. Probabilities are given here to provide an insight into the experimental results and provide a guideline for designing kinematics. Strict stochastic modeling about retaining capabilities will be conducted in the future.

Beads with a higher density than water may fall out of the sub-umbrella region or lag behind the robot due to the gravity, in addition to Mechanisms-1 and 2. This is classified as the escape Mechanism-3. Mechanism-3 is discussed in Supplementary Note 12 and is not included here.

**Materials used in the demonstrated four tasks**. In selective transportation experiments, we use two kinds of polystyrene beads with different sizes (Polysciences, Inc.). The diameters of the large beads are 965–1015 µm, and the diameters of the small beads are 500–600 µm. Both the large and small beads have the same densities (1.05 g·cc⁻¹). In burrowing experiments, the fine beads are 200–300 µm, density of 1.05 g·cc⁻¹ (Polysciences, Inc.). The target objects to be searched are large beads painted with black ink (diameter: 965–1015 µm, density: 1.05 g·cc⁻¹, Polysciences, Inc.). In local mixing experiments, the dyes used for demonstrations are food dyes (Bakeryteam, GmbH). In chemical path generation experiments, we use fluorescein sodium as the chemical to be distributed (Fisher Scientific U.K., Ltd.).

**Statistical method and normalization**. Because of the limited sample size, the two-sided Wilcoxon rank sum test is applied to examine the statistical significance. The test is conducted between the values of Mode A and the values of the other four basic swimming modes. Asterisks are used to denote the statistically significant difference. *, **, and *** denote $P \le 0.05$, $P \le 0.01$, and $P \le 0.001$, respectively.

To easily visualize the difference between Mode A and other modes, the experimental data are normalized. The values of each data set are first divided by the average value of Mode A and then subtract 1. Therefore, a positive value indicates the performance of the corresponding swimming mode outperforms Mode A, while a negative value indicates the corresponding swimming mode underperforms Mode A.

## Data availability

All data generated or analyzed during this study are included in the published article and its Supplementary Information, and are available from the corresponding author on reasonable request.

## Code availability

The MATLAB codes used in this study are available from the corresponding author on reasonable request.

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

## Acknowledgements
We thank all members of the Physical Intelligence Department at the Max Planck Institute for Intelligent Systems for their comments. We also thank J. H. Costello for helpful discussions. This work is funded by the Max Planck Society.

## Author contributions
M.S., Z.R., W.H. and X.D. proposed and designed the research. Z.R., W.H., and X.D. performed all experiments. Z.R., W.H. and X.D. developed the theoretical models and performed the simulations. The experimental data were analyzed by Z.R., W.H. and X.D. All authors wrote the paper and participated in discussions.

## Additional information

**Competing interests:** The authors declare no competing interests.

