## [Peer Review File · Nature Communications]

Reviewers' comments:

Reviewer #1 (Remarks to the Author):

The article presents an interesting and exciting swimming robot design. The robot is well-designed and has very interesting properties. The design of the robot and its performance are good contributions to the state-of-the-art. The manuscript has several places to be improved though. Several design and control choices require further explanation. In particular, the shape and deforming pattern of the "fins" of the jellyfish, and the actuating amplitudes and frequencies are very important, and not sufficiently described and analyzed in the article.

Specific comments:

1. How the structure and stiffness of the fins and body affects the fins deformation and the speed of swimming. What happens to the free edge of the lappet during the actuation? There are many joints and pads in the lappets, what's the design mechanism of the geometry and material properties.
2. How close the actuating modes (the amplitudes, frequency and thrust force) of the robot are compared to real animals (jellyfish). Real jellyfish has a bell shape structure for the propulsion. In this article the robot has eight separated "fins" instead of the bell shape. Is there any designing principles to illustrate this difference?
3. Electromagnetic coil is used to generate driving force to the robot. Is that possible to estimate the input power of this system? What's the efficient of the robot during swimming and control?
4. The robot shows up-down swimming and steering. The electromagnetic coil is fixed outside, what's the limit of control freedom of this system? Is there any method to improve the control performances?
5. It's quite interesting to show the movements of small and big beads (in the sup video). What's the relation between the radius of the bead can be raised and the characteristic length of the robot?
6. Is that possible to estimate the fatigue and life time of this robot? What are the failure modes during the actuation and control? To analyze those modes may help the further design and practical applications.

Reviewer #2 (Remarks to the Author):

The authors have detailed the work of a magnetically actuated mini-robot inspired by jellyfish ephyra. They have detailed the different locomotion modes of the robot and also the other capabilities of the robot such as burrowing, mixing, and manipulation. While there have been other robots in the past inspired by ephyra, this manuscript does have several noteworthy novel aspects, particularly the simultaneous capability for transporting objects while simultaneously swimming. Overall, the authors have done a thorough job in the presentation, development, and evaluation of the manuscript. The supplemental videos are professionally made and support the scientific development well. I am pleased to recommend this article for publication pending minor revisions.

The probabilistic estimate of the particle transportation is based on idealized assumptions. It is not made clear if this probability estimation will scale to many different kinds of particles, nonhomogeneous, irregularly shaped, etc, that would likely be encountered in realistic environments. Perhaps it is too early for this kind of model given the amount of supporting data available.

Line 232: authors state that citation 1 uses pneumatic actuators. However, actuators in that cited paper appear to be filled with seawater. Perhaps hydraulic would be a more accurate descriptor for the actuators.

At first mention of $v_{\text{drift}}(s,t)$ it is not clear what 's' represents.

In extended data Fig. 5, would be interesting to see experimental data compared to the predicted values from the model.

The manuscript was generally well written, but needs to be spell-checked; Typos: 'burry' should be bury, 'molded' should be modeled, 'predicated' should be 'predicted'

Reviewer #3 (Remarks to the Author):

The study presents a jellyfish-like millirobot featuring magnetic composite elastomer lappets which are actuated by means of a time-varying magnetic field. The lappets are connected to elastomeric pads suitably designed to break stroke symmetry between the contraction and the recovery phase, as functional to achieve a net movement per actuation cycle. The millirobot was experimentally tested in a $100 \times 60 \times 40$ mm fluid domain, possibly containing small (neutrally buoyant) beads also observed by a particle-image-velocimetry setup. For most of the experiments, the magnetic induction (max 30mT) was produced by a couple of Helmholtz coils, so that the millirobot was tested in a single swimming direction (the vertical one). For a few experiments, an additional couple of coils were used to demonstrate steering (in a plane).

The authors firstly demonstrated 5 base actuation modes (whence 5 swimming modes). Each mode was determined by a time-profile for magnetic induction control. For each mode the authors obtained experimental results (some were also interpreted based on complementary models). They were assessed as regards the propulsion performance and the possibility to collect/retain the beads, essentially by means of non-contact actions mediated by the stroke-induced flow field (both vorticity and drift effects were considered). Building on the lessons learnt from the base modes, the authors then showed the possibility to selectively transport smaller/larger beads, to hide the millirobot by making it burrow into a layer of beads, to locally mix the fluid, and to leave a chemical trace by swimming along a path (even curved).

The methodology is sound, the presentation style is good, and the visual material is of very high quality. In light of the above resume, the title (Multi-functional soft-bodied jellyfish like swimming) well represents the content of the study. In the essence, the core ingredients are magnetic actuation and soft-material based flapping. The level of novelty is good, although the related magnetic actuation technology is classical and fluid-mediated non-contact bead manipulation was studied in several microrobotics papers.

It seems that the authors did not address the well-known (adverse) scaling effects of magnetic actuation with distance, which can inhibit the function of the proposed system in contexts (even slightly) different from the used setup. Also the relevance of visual cues for robot control is overlooked. Furthermore, it is not clear whether the observed results could be extended in a clear way to additional working conditions, also owing to boundary effects (already pointed in the methods), or unsteady flow effects.

The millirobot takes inspiration from a real jellyfish (ephyra). Its size is comparable. The authors also claimed a stronger fluidic similarity based on the Reynolds number (in the SI). However, the timescales of the real and artificial jellyfish seem to be very different. The flapping frequency of the robot seems to be higher than that of the biological model (the authors could have also considered the related non-dimensional number), so that the comparison in fig.1c is nice in terms of trends yet its meaning seems to be obfuscated by the underlying normalization. For the same reason, all the many links appearing in the text in between the robot and the real jellyfish seem to be hardly justified. Moreover, simultaneous parallels to other biological models (e.g. for burrowing or camouflage) seem to further hamper any strong links to biology. Furthermore, the claim of a bioinspired approach is also

weakened towards the end of the main text, when it is stated that the capability of achieving multi-functional behaviour is essentially based on kinematics, because lappet flapping is intrinsically related to the interplay between geometrical and material properties of the soft robot (up to fluid properties). Moreover (yet still in connection with the above points), the advantage of using the proposed millirobot for better understanding the effects of pollutants and environmental conditions on morphology and kinematics changes of ephyra does not seem clear.

The authors are also encouraged to further proofread for some typos (caption of fig.2, supplementary movie S1, etc.). Eq.(3) should be checked, as well as the definition of the plotted escape probabilities (are they normalized with p_{out} ? If not, where does the latter step in?)

Response to Reviewers

We thank the reviewers and the editor for their helpful comments. We have fully addressed all of these comments in the revised manuscript. Please refer to the specific changes described in the following text. The comments from referees are reprinted in blue Arial, and our responses are marked in black Times New Roman. The changes made to the original manuscript are highlighted in yellow.

Reviewer 1:

The article presents a interesting and exciting swimming robot design. The robot is well-designed and has very interesting properties. The design of the robot and its performance are good contributions to the state-of-the-art. The manuscript has several place to be improved though. Several design and control choices require further explanation. In particular, the shape and deforming pattern of the “fins” of the jellyfish, and the actuating amplitudes and frequencies are very important, and not sufficiently described and analyzed in the article.

Response: We thank the reviewer for the kind comments and suggestions. In the next response to your specific comment #1, we’ll discuss the shape and deforming pattern of the “fins”. Before that, we first discuss the actuating amplitudes and frequencies here.

We agree with the reviewer’s comment that “the actuating amplitudes and frequencies are very important”. In fact, the efforts of prescribing the Mode A, B1, B2, B3, and C are to research these two factors.

Instead of frequency (f), we decide to discuss its inverse, the beating period $T = 1/f$, which is the sum of three individual phases ($t_C+t_R+t_G$). We implement such approach because our research goal is to understand the interaction between the robot’s soft body and incurred fluidic flows due to the robot’s body motion, and utilize such physical interaction for predation-inspired object manipulation capability of the robot, and our specific research methodology is to understand the function of each phase to different performance indices (summarized in Tables 1 and 2). Therefore, being able to tune the duration of each phase independently is crucial (see “**Methods: Kinematics of five main swimming modes**” and **SI section S2.3** in current submission). In contrast, tuning the frequency while maintaining the waveform would change phase durations altogether, making it hard to enforce our research methodology.

When investigating the amplitudes, we focus on the actual beating amplitude of the lappets ($\theta_R - \theta_C$), rather than the \mathbf{B} field amplitude which drives this beating amplitude. The reason is, again, that we want to investigate the interaction between the robot’s soft body and incurred fluidic flows due to the robot’s body motion, and utilize such physical interaction for predation-inspired object manipulation capability of the robot. And we believe such interaction, *i.e.* how the robot should swim for different functions, is independent of \mathbf{B} field and is more phenomenal, as it’s transferrable

to other miniature swimming robot. Therefore, the beating amplitude, which directly correlates to the incurred fluidic flow, is more informational than B field amplitude which only influences indirectly. In the discussion, this specific investigation on beating amplitude is carried out by contrasting the Mode C to the Mode A. To elaborate the investigation more in the current submission, we also add a new discussion with experiment results and modeling to Fig. S10c. Please kindly see **paragraph 8 in SI section S3** of the current submission for the following discussions:

Fig. S10: b-e, Variation of the propulsion performance when changing the recovery angle (θ_R).

“We vary θ_R in Fig. S10c. Specifically, we keep $t_C = 0.13$ s, $t_R = 0.27$ s, $t_G = 0$ s, $\theta_C = 0.6$ rad, which are the same as Mode A, and vary θ_R from 0.6 rad to 2.7 rad. There should exist an optimal value for the beating amplitude to induce the highest swimming velocity. If the beating amplitude is too small, the thrust produced during contraction decreases. While if the beating amplitude is too large, the drag produced during recovery increases. Both of these effects can slow down the robot.”

At last, to compensate the above discussion and provide a picture on the complex effects of solely varying the amplitudes and frequencies (while maintaining the waveform), the first submission, in fact, has provided a frequency response measurement of the whole lappets under different amplitudes (Extended Data Fig. 2b in the first submission). In the current submission, it is discussed in more details and moved to **Paragraph 2 in “Methods: Passive lappets”** and Fig. S2b. The figures and texts are shown below:

Fig. S2: b, The “cut-off frequency” of the lappet. (i) Sinusoidal magnetic fields with different amplitudes and frequencies are applied. Five measurements are conducted at each data point. Linear least square regression is used to find the “cut-off frequency”. The horizontal dashed lines indicate the beating amplitude that is 0.707 times that achieved under 0.5 Hz. (ii) The “cut-off frequency” under the sinusoidal magnetic fields with different amplitudes. See “**Methods: Passive lappets**” for the definition of the “cut-off frequency”.

“With the robot center pinned on a pillar (restricting only the vertical translation), the frequency response of the whole lappet, including both magnetic lappet and passive lappet, is tested under sinusoidal B_y fields of three different magnitudes (10, 20, and 30 mT) and different frequencies (0.5 ~ 30 Hz) (see Fig. S2b). Five measurements are conducted for each case. In this report, we borrow the concept of “cut-off frequency” from the linear system to clarify the relation between the actuating magnetic field (magnitude and frequency) and the resultant beating amplitude. We define the “cut-off frequency” of the lappet to be the frequency at which the beating amplitude, $\theta_R - \theta_C$, is 0.707 times that achieved under 0.5 Hz. Linear least square regression is used to obtain the relation between the actuation frequency and the beating amplitude and to find out the “cut-off frequency” (Fig. S2b-i). In Fig. S2b-ii, we show that the “cut-off frequency” of the lappet can be increased by increasing the B field magnitude. In the future, the non-linearity in frequency response (e.g., the frequency response observed at 10 mT) will be investigated.”

Note that **ii** in the above figure is slightly different from that in the first submission (Extended Data Fig. 2b in the first submission) as more experimental data points are added to generate the estimated cut-off frequency. The change doesn't alter any conclusion, however.

Specific comments :

#1. How the structure and stiffness of the fins and body affects the fins deformation and the speed of swimming. what happens to the free edge of the lappet during the actuation? There are many joints and pads in the lappets, what's the design mechanism of the geometry and material properties.

Response: To address the reviewer's concern, we've added a whole new section in **SI section S4** (Also supported by Fig. S11~ Fig. S15). The discussion mainly relies on different finite element simulations to explain the impacts of varying different design parameters. As the robot consists of an active magnetic composite elastomer core (*i.e.*, body) and eight passive lappets (*i.e.*, fins), we divide the discussion into two parts:

Section S4.1 We discuss **the structure and stiffness of the magnetic composite elastomer core**. Specifically, we focus on the thickness (h), Young's modulus (E), and the magnetization magnitude ($|m|$) and their influences on magnetic core deformation.

Section S4.2 We discuss **the structure and stiffness of the passive lappets (free edge)**. Specifically, we focus on the influence of changing the stiffness of the flexible joints and the length of the pads on kinematics and swimming velocity. In this section, we also discussed the behavior of the free edge of the lappet during the actuation. Generally speaking, the proximal joint bends during recovery while not in contraction. The distal joint bends in both recovery and contraction. Note that the discussion about the "free edge" can be found in the main text of the first submission (**Paragraph 2 in the main text** and Extended Data Fig. 2 of the first submission). In the current submission, it is at **paragraph 1 in "Results: Design and swimming behavior of the ephyra-inspired swimming soft millirobot"** and Fig. S2.

In summary, the simulation results show that our robot design is suboptimal in terms of propulsion velocity. Of course, such a design mechanism only provides the first-step guideline to produce optimized jellyfish robot for different purposes in the future. More detailed design mechanisms should be investigated in the future.

#2. How close the actuating modes (the amplitudes, frequency and thrust force) of the robot are compared to real animals (jellyfish). Real jellyfish has a bell shape structure for the propulsion. In this article the robots has eight separated "fins" instead of the bell shape. Is there any designing principles to illustrate this difference?

Response: Through the manuscript, we show our similarities to real ephyra from the following 9 aspects:

1. **Morphology:** Note that the morphology of the ephyra jellyfish is not continuous. And this can be found in the previous biological study¹. In fact, at their early developmental stage, the ephyra jellyfish possess eight lappets with deep clefts in between². The design of the "eight separated fins" is to capture such characteristic. We have mentioned this point in the paper as (**Paragraph 2 in the main text** of the previous submission, **Paragraph 1 in "Results: Design**

and swimming behavior of the ephyra-inspired swimming soft millirobot” of the current submission):

“Scyphomedusae ephyra (diameter: 1 ~ 10 mm) is characterized by its incomplete bell (Fig. 1a-i) and lappet paddling-based propulsion^{23,25,26,28.}”

Please also refer to Fig. 1a-i of the first submission and Fig. 1a-i of the current submission for illustration.

- 2. Bell fineness:** The comparison of the beating amplitude is shown in Fig. 1c by a parameter called ‘bell fineness’. The bell fineness is defined as the ratio between the bell height and the bell diameter and is commonly used in biological studies of real jellyfish to indicate the alteration of the bell shape in one pulsation cycle^{2,3}. Fig. 1c shows the bell fineness of the robot is very close to the real animal. Please also refer to **SI section S2.4** of the current submission for more information (**SI section S2.4** of the first submission).
- 3. Beating frequencies:** The beating frequency of Mode-A (control group, biomimetic kinematics) is 2.5 Hz, which is in fact prescribed based on the biological observations. Please refer to Fig. 3A-D in Higgins et al³ and Fig. 5 in Feitl et al² for details. Although the relevant information can be found in the first submission (Extended Data Fig. 4d in the first submission), we have pointed this out more clearly in current submission (**paragraph 2 in “Results: Design and swimming behavior of the ephyra-inspired swimming soft millirobot”** of the current submission):

“Keeping the beating frequency (2.5 Hz) and Reynolds number of the robot body ($Re_B = 7 \sim 95$) similar to an ephyra^{25,27,28}, the robot can capture the typical flow patterns of its biological counterpart^{23,30.}”

- 4. Swimming velocity (thrust force):** The velocity of the robot swimming at Mode-A (3.95 mm/s, Fig. 2c) is at the same magnitude of a continuously swimming ephyra as reported in Sullivan et al⁴ (average swimming speed: 2.69 ± 0.98 mms⁻¹). Since the sizes of the ephyrae studied in Sullivan et al⁴ range from 6 to 8 mm in diameter, which is close to the size of our robot (5.6 mm), the thrust force produced by the robot is also at the same magnitude as an ephyra.
- 5. Kinematics profile:** When the robot swims with Mode-A, the lappets can achieve a larger wetted area during contraction than during recovery (see Fig. 2a-ii, Fig. S2c). This feature is in accordance with the real animal (please refer to **SI Section S1** and Fig. S4 in the current submission, **SI Section S1** and Fig. S1 in the first submission).
- 6. Lappet velocity:** When the robot swims with Mode-A, the variation of the normalized lappet velocity in one pulsation (see Fig. 1c) is in accordance with the real animal². The reason why we normalize it is that the biological data is from a smaller ephyra with a diameter of 3 mm. This has been explained in **SI Section S2.4** in the current submission (**SI Section S2.4** in the first submission).
- 7. Reynolds number:** The robot swims at the Reynolds number ranging from 7 to 95, which matches the data reported in Feitl et al.², in which the *Aurelia aurita* ephyra swims within the

Reynolds number ranging from 1 to 100. Please refer to **SI section S2.5** in the current submission for detail (**SI section S2.5** in the first submission).

8. **Wake flow structure:** The robot induces similar wake flow structures (see Fig. 1b) as a real ephyra jellyfish. Please refer to Fig. 3 in Nawroth et al.⁵ and the Fig. 1 in Nawroth et al.⁶ for the wake flow structures observed from the real animals.
9. **Prey trajectories:** The robot has a similar object capture ability (see Fig. 1d) as a real ephyra jellyfish. Please also refer to Fig. 9 of Higgins et al.³ for the prey capture behavior in real ephyra.

At last, it should be noted that the BC (boundary coverage) rate of the real jellyfish is not captured sufficiently. Please refer to **SI section S2.7** of the current submission, **SI section S2.7** of the first submission. However:

1. This work mainly targets for realizing different functions inspired by the predation behavior of the ephyra jellyfish. The discrepancy between the robot and the real animal is acceptable if the robot can generate similar flow structures and achieve different robotic tasks.
2. A full mimic of the real animal is certainly attractive. However, simple robotic platforms which can capture different aspects of the real animals have been widely used to investigate the biomechanical questions^{7,8}.
3. Moreover, we further demonstrate that with the same fabrication and control strategies, we can develop a robotic jellyfish that matches the real animal better in morphology (Fig. S7). This design will be further optimized in the future.

Fig. S7: The robotic jellyfish with a larger boundary coverage (BC) rate. a, The BC rate of the newly-designed robot can reach 60.87%. b, The motion sequence of one beating cycle.

In summary, the robotic jellyfish can capture nine aspects of the real ephyra jellyfish. The fabrication and control strategies proposed can also be optimized further in the future depending on the scientific topic being explored.

#3. Electromagnetic coil is used to generate driving force to the robot. Is that possible to estimate the input power of this system? What's the efficiency of the robot during swimming and control?

Response: We agree that the efficiency is important. Instead of estimating the total input power of the whole coil system, here we focus more on the swimming efficiency of the robot itself (shown in the Fig. 2c), as our research goal is to study the interaction between the robot's soft body and incurred fluidic flows due to the robot's body motion, and utilize such physical interaction for predation-inspired object manipulation capability of the robot (Please check "**Abstract**" and "**Introduction**" in the current submission and "**Abstract**" in the first submission). And we believe such interaction (*i.e.* how the robot should swim for different functions) is independent of \mathbf{B} field and is more phenomenal, as it is transferrable to other miniature swimming robots.

Specifically, we choose to use the cost of transport (COT) as the metrics to quantify the energy efficiency of the robot. The energy input to the robot is calculated by integrating the work done by the distributed magnetic torque acting on the robot body (**SI section S2.6** of the current submission, **SI section S2.6** in the first submission). The COT obtained at different swimming modes is in Fig. 2c.

#4. The robot show up-down swimming and steering. The electromagnetic coil is fixed outside, what's the limit of control freedom of this system? Is there any method to improve the control performances?

Response: The independently controllable degree-of-freedom (DOF) of the robot's rigid-body translational and rotational motions depend on the configuration of the electromagnetic coil system. With the current design of the robot, its swimming direction is always parallel to the external magnetic field due to a net magnetic torque. Therefore, with a single pair of fixed Helmholtz electromagnets, we have one control degree-of-freedom for controlling the 1-DOF rigid body translational motion of the robot, *e.g.* swimming "up-down". With two pairs of fixed Helmholtz electromagnets in an orthogonal configuration, we have two control degree-of-freedom for controlling the 1-DOF rigid body translational motion parallel to the external magnetic field and 1-DOF rotational motion, *i.e.*, "swimming" and "steering" in 2D. To improve the control performance, there are several ways:

- (1) We can use more complex stationary or movable electromagnetic coils (*e.g.* eight-coil system^{9,10}) capable of independently controlling a uniform magnetic field in a three-dimensional space. Using the current robot design, we would have full control of its 1-DOF rigid-body translational motion parallel to the external magnetic field as well as 2-DOF rotational motions (yaw and pitch) in 3D.
- (2) The dynamic model (**SI section S3** in the current submission and **SI section S3** in the previous submission) proposed is used to guide the control signal waveform design. However, this model relies on the drag coefficients from the previous paper which may not fully meet our experimental conditions. And it doesn't include the fluid-structure interaction so that the deformation of the robot during swimming cannot be predicted from the model. Therefore, an

improvement to the dynamic model in the future will help us predict the performance of the robot better, which is favorable for building a model-based control system.

- (3) The high-level swimming and steering motion of the robot is currently teleoperated by an operator. To further improve the control performance such as accuracy of path following and robustness to environmental disturbances, closed-loop control algorithms using visual feedback from video cameras or ultrasound sensors can be potentially applied.

We thank the reviewer to point this out. Therefore, we've also added the above discussion to **Paragraph 2 in "Methods: Electromagnetic coil (magnetic actuation) setup and particle image velocimetry (PIV) system"** in the main manuscript. And it reads:

"Currently, the control of the robot is realized by providing an oscillating magnetic field along its body central axis. Therefore, the controllable degree of freedoms (DOFs) of the robot's rigid-body translational and rotational motions depend on the configuration of the electromagnetic coil system. With a single pair of fixed electromagnets, we have one control DOF for controlling the 1-DOF rigid body translational motion (*e.g.*, swimming vertically in characterization experiments). With one more pair of fixed electromagnets, one more controllable DOF for rotational motion can be obtained and steering the robot in 2D can be achieved (*e.g.*, generating S-shaped chemical path). To improve the control performance, we can add more coils to the system^{55,57}, improve the dynamic model (SI section S3), and implement visual feedback control."

#5. Its quite interesting to show the movements of small and big beads (in the sup video). Whats the relation between the radius of the bead can be raised and the characteristic length of the robot?

Response: We thank the reviewer for being interested in this relation. Therefore, we add a new section in SI (SI section S5.1.3: "Investigating the influence of different factors on object manipulation performance") to discuss it. Through FEA simulations, we get the following results (shown in Fig. S19c):

Fig. S19: c, The normalized transporting heights achieved with different beads size and kinematics.

For the horizontal axis, the bead is normalized by the robot body diameter (the characteristic length). In summary, the relation is nonlinear and heavily depends on the specific kinematics being used (for here, Mode D1 and D2). The simulation results agree with the experiment results shown in Fig. 5a and Fig. S18a of the current submission (Fig. 4a and Extended Data Fig. 7a in the first submission) in the sense that Mode D1 is for lifting up large beads (~ 1 mm) while Mode D2 is for lifting up small beads (~ 0.5 mm). If the bead size keeps increasing (~ 2 mm), the robot eventually cannot lift it up, as expected. For more details, please kindly check **SI section S5.1.3 “Investigating different factors on object manipulation performance”**.

#6. Is that possible to estimate the fatigue and life time of this robot? What are the failure modes during the actuation and control? To analyze those mode may help the further design and practical applications.

Response: During our investigation, we observe that the life time of the robot is quite long, and no fatigue, especially for the magnetic composite elastomer itself, has been spotted throughout our study. From our experience, the kinematics of the robot can keep unchanged for several months when actuated with the same control signals. From previous literature, structures made by Ecoflex can maintain its properties after subjecting to cyclic loading of over 10^6 cycles¹¹. However, systematic tests have to be done in the future to quantify the fatigue properties of the material. We clearly point this out in **Paragraph 1 in “Methods: Electromagnetic coil (magnetic actuation) setup and particle image velocimetry (PIV) system”** of the current submission:

“The characterization experiments are all performed in this setup using multiple robots with the same design. No fatigue of the robot has been observed throughout our experiments.”

As to the second question, there are several circumstances that may cause failure during control:

(1) Due to the fabrication error, the net magnetic moment of the magnetic core may not align perfectly along the body central axis. If the net magnetic moment incurred deviates too much from the body central axis, the magnetic core may toddle or even flip under the oscillating B field. Improving the manufacturing accuracy can greatly reduce the occurrence of this circumstance. We’ve discussed this problem in **paragraph 4 in “Methods: Magnetic composite elastomer core”** of the current submission (**paragraph 4 in “Methods: Magnetic composite elastomer core”** of the first submission):

“... the deformation of the elastic core is not perfectly axial symmetric, causing the net magnetic moment deviating from its central axis and inducing the undesired magnetic torque.”

(2) When the direction of the B field makes the robot tilt larger than 45° from the vertical direction, we observed that the robot tends to toddle. This might be due to the self-righting effects caused by the bubble trapped on its top. We’ve discussed this problem in **paragraph 4 in “Methods: Magnetic composite elastomer core”** of the current submission (**paragraph 4 in “Methods: Magnetic composite elastomer core”** of the first submission):

“...this might be due to the bubble, which is trapped on top of the body, providing a self-righting torque, which always rotates the robot central axis back to be vertical. We observe that such phenomenon relieves when the beating frequency increases and beating amplitude reduces as in Supplementary Video S7. Such dynamic response would be further investigated in the future.”

- (3) During the swimming, the robot may contact with the water surface and rupture the bubble. This may happen when the robot swims so fast that the operator cannot react to turn off the coil system timely. There are two potential solutions regarding this problem. First, a visual-feedback control system can be developed to better control the robot's speed and position. Second, a 3D-printed hollow bead can be used to replace the bubble to reduce the overall density of the robot. Such efforts will be further explored in the future.

Reviewer #2:

The authors have detailed the work of a magnetically actuated mini-robot inspired by jellyfish ephyra. They have detailed the different locomotion modes of the robot and also the other capabilities of the robot such as burrowing, mixing, and manipulation. While there have been other robots in the past inspired by ephyra, this manuscript does have several noteworthy novel aspects, particularly the simultaneous capability for transporting objects while simultaneously swimming. Overall, the authors have done a thorough job in the presentation, development, and evaluation of the manuscript. The supplemental videos are professionally made and support the scientific development well. I am pleased to recommend this article for publication pending minor revisions.

Response: We thank the reviewer for the kind comments. Please refer to our response to the specific questions below.

#1. The probabilistic estimate of the particle transportation is based on idealized assumptions. It is not made clear if this probability estimation will scale to many different kinds of particles, nonhomogeneous, irregularly shaped, etc, that would likely be encountered in realistic environments. Perhaps it is too early for this kind of model given the amount of supporting data available.

Response: First, we agree with the reviewer that the probabilistic estimate is limited. It is unclear whether the escaping probabilities we estimate from our experiments (P'_C and P'_R) still strictly hold under more complex conditions. However, as the first work on investigating the non-contact manipulation of the miniature jellyfish robot (or miniature soft robot) in the moderate Re , we are dedicated to an investigation in the idealized and simplified conditions. Indeed, more experiments are needed to find the extent to which the P'_C and P'_R are still valid. And this could be great topics for the investigation in the next phase. (Please also refer to our response to Reviewer #3, comment #3). To make the discussion clearer, we have added the above limitation to the main text (**Paragraph 6 in “Results: The object retaining capability of the five basic swimming modes”**):

“Note that the probabilistic estimate of the bead transportation is based on idealized assumptions. More experiments are needed to find the extent to which the P'_C and P'_R are still valid, *e.g.*, in more complex conditions.”

Second, the process of the beads transportation shown in our experiments (Fig. 3b) is very complicated. The neutrally buoyant beads are randomly seeded in the water and are randomly trapped by the robot. It is hard to find a simple and deterministic rule to predict the trajectory of each bead being trapped. Even if a precise model can be developed to predict whether a bead can be retained or expelled after one beating cycle, it is hard to experimentally test it since all the initial conditions (*e.g.* initial positions and velocities of the beads) needed for the model are hard to be controlled. Since we know little about the whole transporting process, the methods of probability theory are applicable to describe the complex behavior of the system¹². The number of the beating

cycles needed to expel a trapped bead has a geometric distribution if the following assumptions are assumed to be valid:

- 1) Each beating is an independent event;
- 2) Two outcomes are possible after one beating. In our case, a trapped bead can either be expelled or be retained after one beating cycle;
- 3) Probabilities of each outcome hold constant. In our case, the probability of expelling a bead after one beating cycle is P_{out} and the probability of retaining a bead after one beating cycle is $1-P_{out}$.

To make the discussion clearer, we have added the above discussion to the main text (**Paragraph 1 in “Methods: Estimating the escaping probabilities (P'_C and P'_R)**):

“The process of the bead transportation shown in our experiments (Fig. 3b) is very complicated. The neutrally buoyant beads are randomly seeded in the water and are randomly trapped by the robot. It is hard to find a simple and deterministic rule to predict the trajectory of each bead being trapped. Since we know little about the whole transporting process, probabilistic models are applicable to describe the complex behavior of the system¹².”

In fact, the last beating cycle does have an influence on the following beating cycle, since the initial position and velocity of the bead of the following cycle is influenced by the last beating. The assumption that each trial is independent should be further investigated in the future. We have pointed this out in **Paragraph 4 in “Methods: Estimating the escaping probabilities (P'_C and P'_R)”** of the current submission:

“In fact, the last beating cycle does have an influence on the following beating cycle, since the initial position and velocity of the bead of the following cycle are influenced by the last beating. Probabilities are given here to provide an insight into the experimental results and provide a guideline for designing kinematics. Strict stochastic modeling about retaining capabilities will be conducted in the future.”

At last, we indeed have made new efforts to compensate for the insufficiency of the probability estimation in the current submission. Specifically, we have developed new FEA simulations to explain the beads transportation process. Please refer to the new **SI Section S5.1.3 “Investigating the influence of different factors on object manipulation performance”** in our current submission. In summary, the FEA model can capture our current experimental observations in idealized conditions (SI section S5.1.3). However, its prediction power for complex conditions has to be verified by comparing with experiment results, which will be the research in the next phase.

#2. Line 232: authors state that citation 1 uses pneumatic actuators. However, actuators in that cited paper appear to be filled with seawater. Perhaps hydraulic would be a more accurate descriptor for the actuators.

Response: We thank the reviewer for pointing this out. We have already changed the terminology to make it accurate. Moreover, we have also checked other references carefully to make sure they are cited properly.

#3. At first mention of $v_drift(s,t)$ it is not clear what 's' represents.

Response: We apologize for the confusion here. s represents the coordinates along the reference line in the body frame. To make it clearer, we change $v_{drift}(s,t)$ to v_{drift} in the main text. In Fig. 3a-ii, $v_{drift}(s,t)$ is replaced by $v_{drift}(x',t)$ to make sure it agrees with the coordinate frame (SI Section S2.1: “Coordinate systems” in the current submission, SI Section S2.1: “Coordinate systems” in the first submission).

#4. In extended data Fig. 5, would be interesting to see experimental data compared to the predicted values from the model.

Response: As requested, we have conducted a series of new experiments. And Fig. S10 (extended data Fig. 5 in the first submission) shows the comparison of the experimental results and the theoretical predictions. Generally speaking, the trend of the experimental results matches the prediction from the simple dynamic model.

Fig. S10: Impact of changing the kinematic parameters on propulsion performance investigated through a dynamic model. **a**, Each lappet of the robot is modeled as a linear array of elliptical cylinders. **b-e**, Variation of the propulsion performance when changing the key kinematic parameters. The blue curves are the predictions from the simple dynamic model, and the red points indicate the experiment results. $\Delta S = S_C - S_R$ in **d-ii** is the difference between the displacements during the recovery and contraction phases. See SI section S3 for more details.

#5. The manuscript was generally well written, but needs to be spell-checked; Typos: 'burry' should be bury, 'molded' should be modeled, 'predicated' should be 'predicted'

Response: Thanks for pointing these out. We've done several times more careful proofreading and try our best to correct these typos.

Reviewer #3:

The study presents a jellyfish-like millirobot featuring magnetic composite elastomer lappets which are actuated by means of a time-varying magnetic field. The lappets are connected to elastomeric pads suitably designed to break stroke symmetry between the contraction and the recovery phase, as functional to achieve a net movement per actuation cycle. The millirobot was experimentally tested in a 100×60×40mm fluid domain, possibly containing small (neutrally buoyant) beads also observed by a particle-image-velocimetry setup. For most of the experiments, the magnetic induction (max 30mT) was produced by a couple of Helmholtz coils, so that the millirobot was tested in a single swimming direction (the vertical one). For a few experiments, an additional couple of coils were used to demonstrate steering (in a plane).

The authors firstly demonstrated 5 base actuation modes (whence 5 swimming modes). Each mode was determined by a time-profile for magnetic induction control. For each mode the authors obtained experimental results (some were also interpreted based on complementary models). They were assessed as regards the propulsion performance and the possibility to collect/retain the beads, essentially by means of non-contact actions mediated by the stroke-induced flow field (both vorticity and drift effects were considered). Building on the lessons learnt from the base modes, the authors then showed the possibility to selectively transport smaller/larger beads, to hide the millirobot by making it burrow into a layer of beads, to locally mix the fluid, and to leave a chemical trace by swimming along a path (even curved).

The methodology is sound, the presentation style is good, and the visual material is of very high quality. In light of the above resume, the title (Multi-functional soft-bodied jellyfish like swimming) well represents the content of the study. In the essence, the core ingredients are magnetic actuation and soft-material based flapping. The level of novelty is good, although the related magnetic actuation technology is classical and fluid-mediated non-contact bead manipulation was studied in several microrobotics papers.

Response: First, we agree that the related magnetic actuation technology is classical. However, using magnetic composite elastomer to study the interaction between the robot's soft body and incurred fluidic flows due to the robot's body motion, and utilize such physical interaction for predation-inspired object manipulation capability of the robot at moderate Reynolds number, is novel. We choose to use magnetic actuation because of the following reasons:

- (1) Compared to tethered miniaturized actuation methods such as piezoelectric actuator¹³, shape memory composites¹⁴ and dielectric elastomer actuator¹⁵, the magnetic actuation is wireless, fast, and does not induce the drag force coming from the connecting wire.
- (2) Compared to untethered actuation methods such as ultrasound¹⁶, the magnetic field doesn't cause disturbance to the flow field. Therefore, the interaction between the robot body and the incurred flow structures can be better studied.

To make it clearer, we have added the following discussions in **Paragraph 3 in "Introduction"** of the current submission:

“The magnetic composite elastomer is chosen here because it can be actuated and controlled wirelessly and fast by remote magnetic fields, which have minimal effects on the fluidic flow under investigation.”

Second, there are indeed many previous studies on non-contact manipulation. In low Re , many micron-size robots have been proposed to work near the planar solid surfaces and manipulate objects¹⁷⁻²⁷. Moreover, Huang *et al.*²⁸ can achieve three-dimensional object transportation (6 μm beads) using a ~ 20 μm helical microrobot in Stokes flow. In this regime, viscous force dominates and Navier-Stokes equation can be greatly simplified²⁹. In our work, the jellyfish robot is at the millimeter scale. It operates at moderate Re , where both inertial force and viscous force play critical roles³⁰. And no simple analytical solution is available. Therefore, well-controlled experimental work is especially important. To make the reader understand our novelty better, we have added this discussion in **Paragraph 1 in “Introduction”** of the current submission:

“To achieve the object manipulation function, micrometer-size mobile robots operating in the low *Reynolds* number (Re) regime have been proposed to incur controlled viscous fluidic flows to manipulate objects¹⁰⁻²⁰. However, it is unclear whether such approach is applicable in the moderate Re regime, where both inertial and viscous forces play critical roles²¹”

#1. It seems that the authors did not address the well-known (adverse) scaling effects of magnetic actuation with distance, which can inhibit the function of the proposed system in contexts (even slightly) different from the used setup.

Response: We appreciate the reviewer to point out the scaling law. However, we want to emphasize that the main contribution of this paper is to study the interaction between the robot’s soft body and incurred fluidic flows due to the robot’s body motion, and utilize such physical interaction for predation-inspired object manipulation capability of the robot at moderate Re . Such interaction is independent of the \mathbf{B} field and transferrable to jellyfish-like robots made of other smart materials. Therefore, the scaling effects are not the main concern.

It is true that the \mathbf{B} field magnitude scales with the distance between two coils as $\sim \frac{1}{L^3}$ according to Abbott *et al.*³¹. However, we don’t envision too much challenge even in the potential biomedical application we mentioned when introducing the task of selectively transporting beads in **Paragraph 2 in “Results: Four robotic tasks realized by newly-prescribed modes”** of the current submission. This is because all kinematics in this paper are achieved with \mathbf{B} field no more than 30 mT, and currently developed medical devices for human, such as the one proposed by Rahmer *et al.*³², can provide \mathbf{B} field magnitude well beyond 30 mT.

#2. Also the relevance of visual cues for robot control is overlooked.

Response: We agree with the reviewer that the visual cue does play a part in some of our demonstrations. For example, before conducting burrowing tasks, the robot has to move to the hole filled with beads. This process requires the robot to know where the hole locates, and the visual cue does provide such information. In the future, we can implement visual feedback control to

improve the control performance of the robot, as we have discussed at **paragraph 2 in “Methods: Electromagnetic coil (magnetic actuation) setup and particle image velocimetry (PIV) system”** of the current submission:

“Currently, the control of the robot is realized by providing an oscillating magnetic field along its body central axis. Therefore, the controllable degree of freedoms (DOFs) of the robot’s rigid-body translational and rotational motions depend on the configuration of the electromagnetic coil system. With a single pair of fixed electromagnets, we have one control DOF for controlling the 1-DOF rigid body translational motion (*e.g.*, swimming vertically in characterization experiments). With one more pair of fixed electromagnets, one more controllable DOF for rotational motion can be obtained and steering the robot in 2D can be achieved (*e.g.*, generating S-shaped chemical path). To improve the control performance, we can add more coils to the system^{55,57}, improve the dynamic model (SI section S3), and implement visual feedback control.”

However, in this paper, we focus more on studying the interaction between the robot’s soft body and incurred fluidic flows due to the robot’s body motion, and utilize such physical interaction for predation-inspired object manipulation capability of the robot. This interaction doesn’t rely on the visual cue.

#3. Furthermore, it is not clear whether the observed results could be extended in a clear way to additional working conditions, also owing to boundary effects (already pointed in the methods), or unsteady flow effects.

Response: It’s true that the additional and tougher working conditions can complicate the whole question. But, there is no work available about the non-contact manipulation of a jellyfish-like miniature robot at moderate Re even in an idealized working condition, which is a water tank filled with still water in our case. Therefore, as a first step, it’s imperative to answer the question in an idealized condition. Carrying out the experiments in more complicated cases is the task of the next step in the future. Moreover, such simplification is widely used in previous investigations on non-contact object manipulation of the robots at low Re ¹⁷⁻²⁸, and even in biological studies on jellyfish predation⁴. At last, the objects being manipulated are already within the unsteady flow, *i.e.*, the time-varying flow induced by the unsteady locomotion³³. Therefore, unsteady flow effects have already been involved in our discussions.

After understanding the results in the idealized working conditions, it is, of course, interesting to consider more complicated working conditions. As the first step towards such direction, we add two new results shown as following. But an extensive investigation is out the scope of the current manuscript and will be carried out in the future.

(1) We qualitatively demonstrate the strategy used for transporting spherical objects can also be applied to transport objects with irregular shapes through experiments (please also refer to new **SI Section S5.1.3 “Investigating the influence of different factors on object manipulation performance”** and Fig. S20).

Fig. S20: Demonstration of transporting irregular objects. The objects used are made of PDMS and shown in the second row. Mode D1 is used throughout the experiments.

(2) We also conduct a simulation to study whether the tank bottom has influences on object manipulation performance. The results do not see a significant difference in transportation height of the beads (please also refer to new **SI Section S5.1.3 “Investigating the influence of different factors on object manipulation performance”** and Fig. S19d).

Fig. S19: d, The simulation on boundary effect (Tank bottom). The current result does not show significant impacts of the tank bottom on transportation performance.

#4. The millirobot takes inspiration from a real jellyfish (ephyra). Its size is comparable. The authors also claimed a stronger fluidic similarity based on the Reynolds number (in the SI). However, the timescales of the real and artificial jellyfish seem to be very different. The flapping frequency of the robot seems to be higher than that of the biological model (the authors could have also considered the related non-dimensional number), so that the

comparison in fig. 1c is nice in terms of trends yet its meaning seems to be obfuscated by the underlying normalization.

Response: The beating frequency of Mode-A (control group, biomimetic kinematics) of the robot is 2.5 Hz, which is in fact prescribed according to the biological observations. Therefore, the time scale is the same. Please refer to Fig. 3A-D in Higgins et al³ and Fig. 5 in Feitl et al² for detailed information. We have pointed this out in the first submission (Extended Data Fig. 4d), and make it clearer in the current submission (Fig. S5d, **paragraph 2 in “Results: Design and swimming behavior of the ephyra-inspired swimming soft millirobot”**):

“Keeping the beating frequency (2.5 Hz) and Reynolds number of the robot body ($Re_B = 7 \sim 95$) similar to an ephyra^{25,27,28}, the robot can capture the typical flow patterns of its biological counterpart^{23,30}.”

#5. For the same reason, all the many links appearing in the text in between the robot and the real jellyfish seem to be hardly justified.

Response: Please kindly check the response to the comment #4 and the response to Reviewer #1, comment #2, where links between the organisms and robot are established (we show the similarities to real ephyra from 9 aspects).

#6. Moreover, simultaneous parallels to other biological models (e.g. for burrowing or camouflage) seem to further hamper any strong links to biology.

Response: Taking inspirations from a merge of different biological models is also a common method to build bio-inspired robots. For example:

- 1) Morin et al.³⁴ propose a soft machine that can achieve both camouflage and display. The camouflage function is inspired by cephalopods, chameleons, and insects that can dynamically change their body patterns and color for disguise. While the display function is inspired by jellyfish and fireflies who can use bioluminescence for communication.
- 2) Bachmann et al.³⁵ propose a biologically inspired micro-vehicle capable of aerial and terrestrial locomotion by integrating two biologically inspired mechanisms into its design: a compliant wheel-leg terrestrial running gear and chord-wise compliant wings.
- 3) Yu et al.³⁶ propose a robot capable of multimodal locomotion inspired by various amphibian principles of the animals. A hybrid propulsive mechanism coupled with wheel-propeller-fin movements is proposed to integrate fish- or dolphin-like swimming and wheel-based crawling. Its control system is built by taking the inspiration from the speed regulation in *cyprinoids*.

#7. Furthermore, the claim of a bioinspired approach is also weakened towards the end of the main text, when it is stated that the capability of achieving multi-functional behaviour is essentially based on kinematics, because lappet flapping is intrinsically related to the interplay between geometrical and material properties of the soft robot (up to fluid properties).

Response: Like all animals in the fluidic environment³⁷⁻³⁹, the kinematics of the robot is indeed coupled (interplayed) with the fluidic flow (fluid properties). Therefore, this interplay does not weaken the bio-inspiration approach. In fact, taking the advantage of the interplay between the body and the fluid is the bio-inspiration itself, as the ephyra jellyfish also relies on this interplay to predate as shown in Sullivan et al⁴ and Higgins et al³.

Moreover, although the geometrical and material properties in our robot are indeed different from the real animals, the robot can still capture the same interplay as long as the resultant kinematics is similar to that of the real ephyra. This research method of exploiting the fluid-structure interaction, *i.e.*, the interplay between the robot body and the flow structures, to build the robotic platform has even been widely used to go back and study the biological target itself (biomechanics studies). For example, the review paper titled “**Robotics-inspired biology**”⁷ has the following discussion:

“...in reactive model systems (robotic platform), the resultant behaviors or kinematics of movement cannot necessarily be predicted a priori because movement is determined by interaction with the environment (for example via fluid-structure interactions)... If some of the materials used to construct reactive models are flexible and these flexible components interact with the environment, then complex dynamic behaviors can emerge... Reactive model systems enable new insights into biological systems because they may highlight new movement behaviors and emergent dynamics that cannot be predicted from knowledge of organismal morphology and actuation patterns alone...”

As a typical example, biologists use mechanically-actuated flapping foil models to study the influence of the tail shape of the fish on swimming performance⁴⁰. The foils used in the experiments are passive with different flexural stiffness. The midline kinematics of the foil is totally determined by the interplay between the foil and the fluid.

To make the claim towards the end of the main text more accurate, we certainly agree that it would be more accurate to say that the functionality is achieved based on this interaction (interplay), rather than kinematics. For the same reason, in the current submission, we claim our contribution is to understand the interaction between the robot’s soft body and incurred fluidic flows due to the robot’s body motion, and utilize such physical interaction for predation-inspired object manipulation capability of the robot. And we have made the change mentioned by the reviewer to (In **Paragraph 1 in “Discussion”** of the current submission):

“Other jellyfish-like robots could also realize the same multiple functions and tasks if they generate the same proposed lappet kinematics and local flow structures at the same length and time scale, since the incurred flow structures only rely on the interplay between the robot body and the fluid.”

#8. Moreover (yet still in connection with the above points), the advantage of using the proposed millirobot for better understanding the effects of pollutants and environmental conditions on morphology and kinematics changes of ephyra does not seem clear.

Response: As we have discussed in our response to reviewer comment #7, our robotic jellyfish can work as a reactive model to study biological questions. In compared to biological studies, the advantages of using the robotic platforms are^{7,8}:

- 1) They subject to less influence from physiological factors.
- 2) Their motions are repeatable.
- 3) They offer access to variables or quantities that would be difficult to measure on animals.
- 4) They can perform movements that are unnatural or dangerous for animals.
- 5) Their morphology can be systematically changed.

The well-controlled experiments results obtained on the robotic platform can be used to capture important aspects of the real biological process, *e.g.*, the change of the swimmer behaviors on the predation performance. And these results can extrapolate the consequences of the environmental variation on the jellyfish population and the global ecological system.

To make it clearer, we also add the above discussion to **paragraph 2 in “Discussion”** of the current submission:

“Additionally, this robotic platform could also be used as a scientific tool to further study the behaviors of ephyrae due to its various advantages, such as the ability to change the locomotion mode on demand and not being influenced by physiological factors^{52,53}.”

#9. The authors are also encouraged to further proofread for some typos (caption of fig.2, supplementary movie S1, etc.). Eq.(3) should be checked, as well as the definition of the plotted escape probabilities (are they normalized with p_{out} ? If not, where does the latter step in?)

Response: We thank the reviewer for pointing this out. We have carefully proofread the manuscript to address all the typos we found.

Please kindly refer to “**Methods: Estimating the escaping probabilities (P'_C and P'_R)**” of the first submission and in the same section of the current submission, where the meaning and derivation process of P'_C and P'_R have already been discussed.

Moreover, we checked Eq. (3). Indeed, we found a typo. The previous formula is:

$$E_{retain-1}[n] = \lim_{k \rightarrow \infty} P_C \left(\frac{1 - (1 - P_{out})^{k+1}}{P_{out}^2} - \frac{(k+1)(1 - P_{out})^k}{P_{out}} \right)$$

The correct formula is:

$$E_{retain-1} = \lim_{k \rightarrow \infty} P_C \left(\frac{1 - (1 - P_{out})^k}{P_{out}^2} - \frac{k(1 - P_{out})^k}{P_{out}} \right)$$

We want to emphasize this doesn't affect the conclusion. This is because both equations reduce to the same one when $k \rightarrow \infty$.

However, Note that P'_C and P'_R for plotting are not normalized to P_{out} . P_{out} comes into the final expression because the formulation has the sum of a geometrical series. For clarification, the detailed derivation of Equation (3) is provided below:

Firstly, as discussed in “**Methods: Estimating the escaping probabilities (P'_C and P'_R)**”, we assume the number of the beating cycles needed to expel a trapped bead has a geometric distribution:

“As for a neutrally buoyant bead, we assume each swimming cycle of the robot is an independent event that can produce two outcomes: (i) the trapped bead escapes the sub-umbrella region with probability P_{out} ; (ii) the trapped bead is still retained within the sub-umbrella region with probability $1-P_{out}$.”

With the above assumption, we want to ask: How many beating cycles can the robot retain a trapped bead? To answer this question, we can calculate the expected number of the retaining cycles of a trapped bead:

$$E_{retain} = \lim_{k \rightarrow \infty} \sum_{n=1}^k n(1 - P_{out})^{n-1} P_{out}. \quad (9.1)$$

However, we want to further figure out how many beating cycles does a robot need to expel a trapped bead through Mechanism-1 and 2, respectively. Notice that in outcome (i), the escape probability P_{out} can be expressed as $P_{out}=P_C+P_R$ because the beads can escape either through Mechanism-1 with probability P_C (during contraction) or through Mechanism-2 with probability P_R (during recovery). With the above assumptions, Equation (9.1) can be formulated as:

$$\begin{aligned} E_{retain} &= \lim_{k \rightarrow \infty} \sum_{n=1}^k n(1 - P_{out})^{n-1} (P_C + P_R) \\ &= P_C \lim_{k \rightarrow \infty} \sum_{n=1}^k n(1 - P_{out})^{n-1} + P_R \lim_{k \rightarrow \infty} \sum_{n=1}^k n(1 - P_{out})^{n-1} \\ &= E_{retain-1} + E_{retain-2}, \end{aligned} \quad (9.2)$$

where $E_{retain-1}$ and $E_{retain-2}$ are, respectively, the expected number of the retaining cycles of a bead that is expelled through Mechanism-1 and Mechanism-2. We then provides the detailed derivation of $E_{retain-1}$ since $E_{retain-2}$ can be derived with the same procedures. According to equation (9.2), $E_{retain-1}$ can be expressed as:

$$E_{retain-1} = P_C \lim_{k \rightarrow \infty} \sum_{n=1}^k n(1 - P_{out})^{n-1}. \quad (9.3)$$

To further simplify the equation, we let:

$$z = 1 - P_{out}, \quad (9.4)$$

and:

$$S = \sum_{n=1}^k n z^{n-1}. \quad (9.5)$$

By multiplying z on both sides of equation (9.5), we can obtain:

$$zS = \sum_{n=1}^k n z^n. \quad (9.6)$$

By subtracting equation (9.6) from equation (9.5), we can obtain:

$$(1 - z)S = \sum_{n=1}^k z^{n-1} - kz^k. \quad (9.7)$$

From equation (9.7), we can obtain:

$$S = \frac{\sum_{n=1}^k z^{n-1} - kz^k}{1-z} = \frac{1-(1-P_{out})^k}{P_{out}^2} - \frac{k(1-P_{out})^k}{P_{out}}. \quad (9.8)$$

With equation (9.8), equation (9.3) can be written as:

$$E_{retain-1} = P_C \lim_{k \rightarrow \infty} S = \lim_{k \rightarrow \infty} P_C \left(\frac{1-(1-P_{out})^k}{P_{out}^2} - \frac{k(1-P_{out})^k}{P_{out}} \right). \quad (9.9)$$

Equation (9.9) is the equation (3) in the manuscript. When $k \rightarrow \infty$, $(1 - P_{out})^k \rightarrow 0$ because $0 < P_{out} < 1$. Therefore, equation (9.9) can be further simplified as:

$$E_{retain-1} = \frac{P_C}{P_{out}^2}. \quad (9.10)$$

Similarly, we can also obtain the expected number of the retaining cycles of a trapped bead that escapes through Mechanism-2, $E_{retain-2}$:

$$E_{retain-2} = \frac{P_R}{P_{out}^2}. \quad (9.11)$$

Lastly, if we know $E_{retain-1}$ and $E_{retain-2}$, we can calculate P_C and P_R from equations (9.10) and (9.11). However, this is impossible since we cannot do infinite experiments. However, we can calculate P'_C and P'_R as an alternative as we have discussed in **“Methods: Estimating the escaping probabilities (P'_C and P'_R)”**:

“As it is impossible to implement infinite trials of experiments ($k \rightarrow \infty$), theoretically we cannot get the P_C and P_R from the experiment results. However, if we assume the average retaining cycles of the trapped beads from the beads tracing experiments (Fig. 3b-ii) as reasonable estimations to $E_{retain-1}$ and $E_{retain-2}$, then we can obtain probabilities P'_C and P'_R for each swimming mode (Fig. 4g). Therefore, if $P_C = P'_C$ and $P_R = P'_R$, then $E_{retain-1}$ and $E_{retain-II}$ are the expected value for the cycling number recorded in the experiment.”

Reference

- 1 Purcell, J. E. & Angel, D. L. *Jellyfish blooms: New problems and solutions*. (Springer, 2015).
- 2 Feitl, K. E., Millett, A. F., Colin, S. P., Dabiri, J. O. & Costello, J. H. Functional morphology and fluid interactions during early development of the scyphomedusa *Aurelia aurita*. *Biol. Bull.* **217**, 283-291 (2009).
- 3 Higgins III, J., Ford, M. & Costello, J. Transitions in morphology, nematocyst distribution, fluid motions, and prey capture during development of the scyphomedusa *Cyanea capillata*. *Biol. Bull.* **214**, 29-41 (2008).
- 4 Sullivan, B. K., Suchman, C. L. & Costello, J. H. Mechanics of prey selection by ephyrae of the scyphomedusa *Aurelia aurita*. *Mar. Biol.* **130**, 213-222 (1997).
- 5 Nawroth, J. C., Lee, H., Feinberg, A. W., Ripplinger, C. M., McCain, M. L., Grosberg, A., Dabiri, J. O. & Parker, K. K. A tissue-engineered jellyfish with biomimetic propulsion. *Nat. Biotechnol.* **30**, 792-797 (2012).
- 6 Nawroth, J. C. & Dabiri, J. O. Induced drift by a self-propelled swimmer at intermediate Reynolds numbers. *Phys. Fluids* **26**, 091108 (2014).
- 7 Gravish, N. & Lauder, G. V. Robotics-inspired biology. *J. Exp. Biol.* **221**, jeb138438 (2018).
- 8 Ijspeert, A. J. Biorobotics: Using robots to emulate and investigate agile locomotion. *Science* **346**, 196-203 (2014).
- 9 Lum, G. Z., Ye, Z., Dong, X., Marvi, H., Erin, O., Hu, W. & Sitti, M. Shape-programmable magnetic soft matter. *Proc. Natl. Acad. Sci.* **113**, E6007-E6015 (2016).
- 10 Kummer, M. P., Abbott, J. J., Kratochvil, B. E., Borer, R., Sengul, A. & Nelson, B. J. OctoMag: An electromagnetic system for 5-DOF wireless micromanipulation. *IEEE Trans. Robot.* **26**, 1006-1017 (2010).
- 11 Mosadegh, B., Polygerinos, P., Keplinger, C., Wennstedt, S., Shepherd, R. F., Gupta, U., Shim, J., Bertoldi, K., Walsh, C. J. & Whitesides, G. M. Pneumatic networks for soft robotics that actuate rapidly. *Adv. Funct. Mater.* **24**, 2163-2170 (2014).
- 12 Simon, J. *The art of empirical investigation*. (Routledge, 2017).
- 13 Ma, K. Y., Chirarattananon, P., Fuller, S. B. & Wood, R. J. Controlled flight of a biologically inspired, insect-scale robot. *Science* **340**, 603-607 (2013).
- 14 Tolley, M. T., Felton, S. M., Miyashita, S., Aukes, D., Rus, D. & Wood, R. J. Self-folding origami: shape memory composites activated by uniform heating. *Smart Mater. Struct.* **23**, 094006 (2014).
- 15 Acome, E., Mitchell, S., Morrissey, T., Emmett, M., Benjamin, C., King, M., Radakovitz, M. & Keplinger, C. Hydraulically amplified self-healing electrostatic actuators with muscle-like performance. *Science* **359**, 61-65 (2018).
- 16 Ahmed, D., Dillinger, C., Hong, A. & Nelson, B. J. Artificial Acousto-Magnetic Soft Microswimmers. *Adv. Mater. Technol.* **2**, 1700050 (2017).

- 17 Petit, T., Zhang, L., Peyer, K. E., Kratochvil, B. E. & Nelson, B. J. Selective trapping and manipulation of microscale objects using mobile microvortices. *Nano Lett.* **12**, 156-160 (2011).
- 18 Floyd, S., Pawashe, C. & Sitti, M. Two-dimensional contact and noncontact micromanipulation in liquid using an untethered mobile magnetic microrobot. *IEEE Trans. Robot.* **25**, 1332-1342 (2009).
- 19 Peyer, K. E., Zhang, L. & Nelson, B. J. Localized non-contact manipulation using artificial bacterial flagella. *Appl. Phys. Lett.* **99**, 174101 (2011).
- 20 Zhang, L., Petit, T., Peyer, K. E. & Nelson, B. J. Targeted cargo delivery using a rotating nickel nanowire. *Nanomed.-Nanotechnol. Biol. Med.* **8**, 1074-1080 (2012).
- 21 Tung, H. W., Peyer, K. E., Sargent, D. F. & Nelson, B. J. Noncontact manipulation using a transversely magnetized rolling robot. *Appl. Phys. Lett.* **103**, 114101 (2013).
- 22 Ye, Z., Diller, E. & Sitti, M. Micro-manipulation using rotational fluid flows induced by remote magnetic micro-manipulators. *J. Appl. Phys.* **112**, 064912 (2012).
- 23 Hu, W., Fan, Q. & Ohta, A. T. An opto-thermocapillary cell micromanipulator. *Lab Chip* **13**, 2285-2291 (2013).
- 24 Pawashe, C., Floyd, S., Diller, E. & Sitti, M. Two-dimensional autonomous microparticle manipulation strategies for magnetic microrobots in fluidic environments. *IEEE Trans. Robot.* **28**, 467-477 (2012).
- 25 Zhang, L., Peyer, K. E. & Nelson, B. J. Artificial bacterial flagella for micromanipulation. *Lab Chip* **10**, 2203-2215 (2010).
- 26 Ye, Z. & Sitti, M. Dynamic trapping and two-dimensional transport of swimming microorganisms using a rotating magnetic microrobot. *Lab Chip* **14**, 2177-2182 (2014).
- 27 Zhou, Q., Petit, T., Choi, H., Nelson, B. J. & Zhang, L. Dumbbell Fluidic Tweezers for Dynamical Trapping and Selective Transport of Microobjects. *Adv. Funct. Mater.* **27**, 1604571 (2017).
- 28 Huang, T. Y., Qiu, F. M., Tung, H. W., Chen, X. B., Nelson, B. J. & Sakar, M. S. Generating mobile fluidic traps for selective three-dimensional transport of microobjects. *Appl. Phys. Lett.* **105**, 114102 (2014).
- 29 Purcell, E. M. Life at Low Reynolds-Number. *Am. J. Phys.* **45**, 3-11 (1977).
- 30 El Yacoubi, A., Xu, S. & Wang, Z. J. Computational study of the interaction of freely moving particles at intermediate Reynolds numbers. *J. Fluid Mech.* **705**, 134-148 (2012).
- 31 Abbott, J. J., Peyer, K. E., Lagomarsino, M. C., Zhang, L., Dong, L., Kaliakatsos, I. K. & Nelson, B. J. How should microrobots swim? *Int. J. Robot. Res.* **28**, 1434-1447 (2009).
- 32 Rahmer, J., Stehning, C. & Gleich, B. Remote magnetic actuation using a clinical scale system. *PLoS one* **13**, e0193546 (2018).
- 33 Liu, H., Kolomenskiy, D., Nakata, T. & Li, G. Unsteady bio-fluid dynamics in flying and swimming. *Acta Mechanica Sinica* **33**, 663-684 (2017).

- 34 Morin, S. A., Shepherd, R. F., Kwok, S. W., Stokes, A. A., Nemiroski, A. & Whitesides, G. M. Camouflage and display for soft machines. *Science* **337**, 828-832 (2012).
- 35 Bachmann, R. J., Boria, F. J., Vaidyanathan, R., Ifju, P. G. & Quinn, R. D. A biologically inspired micro-vehicle capable of aerial and terrestrial locomotion. *Mech. Mach. Theory* **44**, 513-526 (2009).
- 36 Yu, J., Ding, R., Yang, Q., Tan, M., Wang, W. & Zhang, J. On a bio-inspired amphibious robot capable of multimodal motion. *IEEE-ASME Trans. Mechatron.* **17**, 847-856 (2012).
- 37 Gemmell, B. J., Costello, J. H., Colin, S. P., Stewart, C. J., Dabiri, J. O., Tafti, D. & Priya, S. Passive energy recapture in jellyfish contributes to propulsive advantage over other metazoans. *Proc. Natl. Acad. Sci.* **110**, 17904-17909 (2013).
- 38 Lucas, K. N., Johnson, N., Beaulieu, W. T., Cathcart, E., Tirrell, G., Colin, S. P., Gemmell, B. J., Dabiri, J. O. & Costello, J. H. Bending rules for animal propulsion. *Nat Commun* **5**, 3293 (2014).
- 39 Beal, D., Hover, F., Triantafyllou, M., Liao, J. & Lauder, G. V. Passive propulsion in vortex wakes. *J. Fluid Mech.* **549**, 385-402 (2006).
- 40 Feilich, K. L. & Lauder, G. V. Passive mechanical models of fish caudal fins: effects of shape and stiffness on self-propulsion. *Bioinspir. Biomim.* **10**, 036002 (2015).

REVIEWERS' COMMENTS:

Reviewer #1 (Remarks to the Author):

The article presents a interesting and exciting swimming robot design. The robot is well designed and the investigations of robot are thoroughly carried out. The videos of this work are very clear and inspiring for the further research. The revised manuscripted is improved in both the illustration of experiments and analysis. I am pleased to recommend this article for publication.

Reviewer #2 (Remarks to the Author):

The authors have satisfactorily addressed my comments. It is an interesting work.

Reviewer #3 (Remarks to the Author):

Authors fully addressed the comments and remarks of reviewers, improving the scientific soundness and quality of the manuscript. The work is original and well presented.

Response to Reviewers

We thank the reviewers and the editor for their helpful comments. The comments from referees are reprinted in **blue Arial**, and our responses are marked in black Times New Roman.

Reviewer 1:

The article presents a interesting and exciting swimming robot design. The robot is well designed and the investigations of robot are thoroughly carried out. The videos of this work are very clear and inspiring for the further research. The revised manuscript is improved in both the illustration of experiments and analysis. I am pleased to recommend this article for publication.

We thank the reviewer for the kind comments.

Reviewer 2:

The authors have satisfactorily addressed my comments. It is an interesting work.

We thank the reviewer for the kind comments.

Reviewer 3:

Authors fully addressed the comments and remarks of reviewers, improving the scientific soundness and quality of the manuscript. The work is original and well presented.

We thank the reviewer for the kind comments.